# Cell-specific alterations in *Pitx1* regulatory landscape activation caused by the loss of a single enhancer

Raquel Rouco [1,2,5], Olimpia Bompadre [1,2,5], Antonella Rauseo[1,2], Olivier Fazio[3], Rodrigue Peraldi[1,2,4], Fabrizio Thorel[3] & Guillaume Andrey [1,2✉]

Developmental genes are frequently controlled by multiple enhancers sharing similar specificities. As a result, deletions of such regulatory elements have often failed to reveal their full function. Here, we use the *Pitx1* testbed locus to characterize in detail the regulatory and cellular identity alterations following the deletion of one of its enhancers (*Pen*). By combining single cell transcriptomics and an in-embryo cell tracing approach, we observe an increased fraction of *Pitx1* non/low-expressing cells and a decreased fraction of *Pitx1* high-expressing cells. We find that the over-representation of *Pitx1* non/low-expressing cells originates from a failure of the *Pitx1* locus to coordinate enhancer activities and 3D chromatin changes. This locus *mis*-activation induces a localized heterochrony and a concurrent loss of irregular connective tissue, eventually leading to a clubfoot phenotype. This data suggests that, in some cases, redundant enhancers may be used to locally enforce a robust activation of their host regulatory landscapes.

[1] Department of Genetic Medicine and Development, Faculty of Medicine, University of Geneva, Geneva, Switzerland. [2] Institute of Genetics and Genomics in Geneva (iGE3), University of Geneva, Geneva, Switzerland. [3] Transgenesis Core Facility, Faculty of Medicine, University of Geneva, Geneva, Switzerland. [4]Present address: Genetics and Development Research Unit, Institut de Recherches Cliniques de Montréal, Montréal, QC, Canada. [5]These authors contributed equally: Raquel Rouco, Olimpia Bompadre. ✉email: guillaume.andrey@unige.ch

Alteration in the enhancer composition of regulatory landscapes at developmental genes can lead to pathologies by modifying the dosage and/or distribution of gene transcription[1]. Indeed, over the past years, losses of single regulatory units within complex and partially redundant regulatory landscapes were shown to have clear phenotypical outcomes despite inducing only partial decreases in average transcription[2–5]. As the alterations in the regulatory mechanisms following enhancer deletion have mostly been characterized using bulk tissue analysis, it has been difficult to determine the cell-specific variability behind the loss of expression that accounts for phenotypes. In order to understand the precise molecular origin of these phenotypes, it is therefore essential to characterize how a single enhancer contributes to the activation of entire regulatory landscapes in specific cell populations.

An effective model system to address these unsolved questions is the limb bud, where organogenesis requires a tight control of gene transcription to achieve correct patterning[6]. Critical to this is *Pitx1*, a transcription factor coding gene that is normally expressed in developing hindlimb buds, but not forelimbs, which channels the limb development program to differentiate into a leg[7–9]. Consequently, forelimb *Pitx1* gain-of-function can induce an arm-to-leg transformation, featured by the appearance of an ectopic patella as well as complex changes in the muscular and tendons wiring[10,11]. In contrast, *Pitx1* knock out has been shown to induce partial leg-to-arm transformations with the disappearance of the patella as well as long bone dysplasia and polydactyly[10,12,13]. Unexpectedly, bulk transcriptomics strategies have only revealed marginal downstream gene expression changes upon *Pitx1* loss, suggesting that an interplay between these changes and the growth rate of limb cell subpopulations collectively result in the various phenotypes[8,10,11,13,14].

As for many developmental genes, several enhancers coordinate *Pitx1* expression in hindlimbs and other tissues. So far, four enhancers have been identified in mammals: *PelB* which drives a distal reporter pattern in hindlimbs, *PDE* that drives expression in the mandibular arch, *RA4* that can drive reporters in a subset of fore- and hindlimb cells and finally, *Pen*, a mesenchymal enhancer that drives expression in both fore- and hindlimbs[11,15,16]. Only the activity of *Pen* was so far shown to strongly contribute to *Pitx1* function in the hindlimb as its deletion leads to a 35–50% reduction of *Pitx1* expression[11]. The deletion of *Pen* has no impact on bone length or digit numbers but induces a partially penetrant clubfoot phenotype, similar to the one observed in mice and humans upon *Pitx1* haploinsufficiency[11,14]. One particularity to the *Pitx1* locus is that it establishes fundamentally different 3D chromatin conformations in transcriptionally active hindlimbs and inactive forelimbs. In active hindlimbs, *Pitx1* forms chromatin interactions with cognate *cis*-regulatory regions spread over 400 kbs, including *Pen* as well as *PDE, RA4*, and *PelB*. In contrast, in inactive forelimbs these interactions are absent and the *Pitx1* gene forms a contact with the polycomb-repressed gene *Neurog1*[11].

In this work, we use a combination of single cell transcriptomics (scRNA-seq), a fluorescent cell-tracing approach and genomic technologies to define the contribution of a single enhancer (*Pen*) in establishing the epigenetically- and structurally-active *Pitx1* regulatory landscape. Moreover, we investigate whether changes in enhancer activities or 3D structure fundamentally associate with transcription or if those can be functionally disconnected of the transcriptional process. Finally, we assess if *Pitx1* expression is homogenous across limb cell populations and if distinct expression levels rely on different enhancer repertoires or, alternatively, in progressive changes in cis-regulatory landscape activities.

## Results

**Two approaches to track *Pitx1* activities suggest a bimodal *cis*-regulatory behavior**. In order to characterise transcriptional, chromatin and structural changes following the *Pen* enhancer deletion, we combined genetic manipulation of the *Pitx1* locus with scRNA-seq and chromatin analysis of sorted limb cell populations. Both approaches enabled characterization of complementary features of gene transcriptional regulation following alterations of the *cis*-regulatory landscape.

First, to define the hindlimb cell types that are expressing *Pitx1* and to assess how the *Pen* enhancer regulates its expression in these cells, we generated single-cell preparations from wildtype (*Pitx1*$^{+/+}$) fore- and hindlimb buds as well as *Pen* enhancer deleted (*Pitx1*$^{Pen-/Pen-}$) or *Pitx1* knocked-out (*Pitx1*$^{-/-}$) hindlimbs (Fig. 1A). We performed 10× genomics in duplicates from E12.5 limb buds as these correspond to a transition stage between patterning and cell-differentiation phases. By performing unsupervised clustering of all the wildtype and mutant single cell transcriptomic datasets, we identified five clusters, to which all the dataset contributed, corresponding to the main populations of the limb: one mesenchymal cluster (*Prrx1+, Prrx2+, Twist1+*; 89% of the cells) and four non-mesenchymal satellite clusters including muscle (*Myod1+, Ttn+; Myh3+*; 4% of the cells), epithelium (*Wnt6+, Krt14+*; 5% of the cells), endothelium (*Cdh5+, Cldn5+*; 1% of the cells) and one immune cell cluster (*C1qa+, Ccr1+*; 1% of the cells) (Fig. 1B, Fig. S1, Supplementary Dataset S1). Yet, as *Pitx1* is mostly expressed in the hindlimb mesenchymal cluster, further analyses were performed only in these cells (Fig. 1C).

In parallel, we devised a fluorescent reporter system to track the regulatory activities of the *Pitx1* locus in hindlimbs (Fig. 1D). Specifically, we first established a reporter line (*Pitx1*$^{GFP}$) by homozygously integrating a regulatory sensor cassette, constituted of a minimal *β-globin* promoter and an EGFP reporter gene, 2 kb upstream of the *Pitx1* promoter in mouse embryonic stem cells (mESCs). These cells were re-targeted to obtain a homozygous deletion of the *Pen* enhancer (*Pitx1*$^{GFP;ΔPen}$). Embryos were then derived from the mESCs via tetraploid complementation[17]. Conventional and light sheet imaging of *Pitx1*$^{GFP}$ embryos showed that the reporter was expressed in all *Pitx1* expression domains including the pituitary gland, the mandible, the genital tubercle and the hindlimbs (Figs. 1E, S2A, Supplementary Video S1)[13,18,19]. In order to investigate potential alterations of gene expression following the EGFP transgene integration, we produced E12.5 bulk hindlimb transcriptomes in both *Pitx1*$^{+/+}$ and *Pitx1*$^{GFP}$. Here, we did not observe a change in *Pitx1* expression suggesting that the insertion of the EGFP transgene did not alter *Pitx1* regulation (Supplementary Dataset S2).

We then FACS sorted GFP+ and GFP− cells from E12.5 *Pitx1*$^{GFP}$ hindlimbs and processed cells for RNA-seq, ChIP-seq and Capture-HiC (C-HiC) (Figs. 1F–G, S2B, C). We found that 8% of the cells in *Pitx1*$^{GFP}$ hindlimbs displayed no EGFP signal, thereby suggesting that the majority of hindlimb cells possesses an active *Pitx1* regulatory landscape. We next compared the transcriptome of GFP+ and GFP− cells and observed a 40-fold enrichment for *Pitx1* expression in GFP+ cells, validating the *Pitx1*$^{GFP}$ allele to track the *Pitx1* regulatory landscape activities (Figs. 1F, S3A, Supplementary Dataset S3). As expected from our scRNA-seq analyses, we found that GFP+/*Pitx1*+ cells were enriched for limb mesenchymal derivatives markers (*Prrx1, Prrx2, Twist1, Sox9, Col2a1, Col3a1, Lum*) and that GFP−/*Pitx1*− were enriched for markers of non-mesenchymal satellite clusters including muscle (*Myod1, Ttn*), epithelium (*Wnt6, Krt15*), endothelium (*Cdh5, Cldn5*) and immune cells (*C1qa, Ccr1*) (Fig. S3B, Supplementary Dataset S3). Yet, the enrichment of

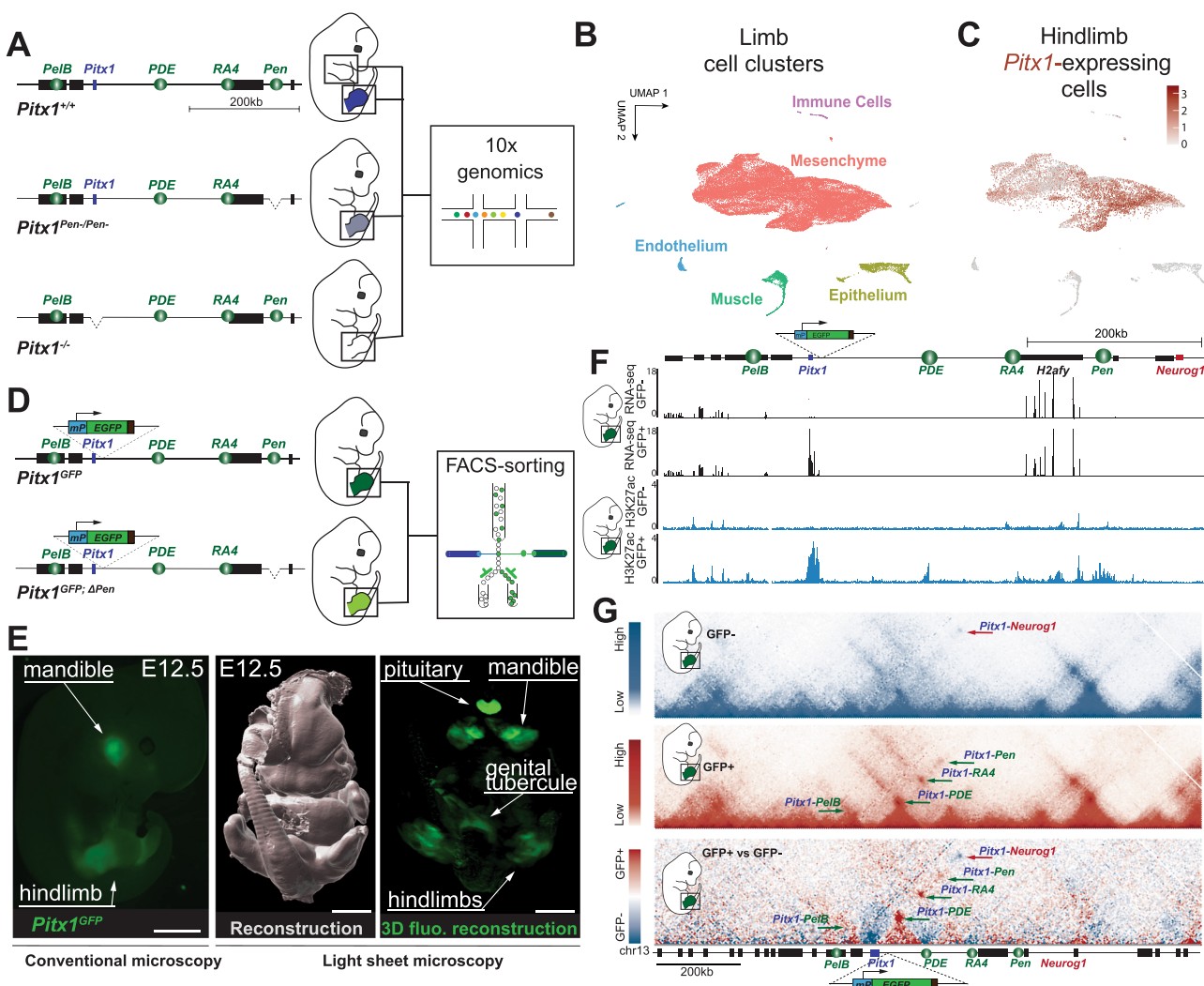

**Fig. 1 Experimental setup, single cell clustering and regulatory sensor. A** $Pitx1^{+/+}$, $Pitx1^{Pen-/Pen-}$, and $Pitx1^{-/-}$ transgenic E12.5 embryos were obtained by tetraploid complementation and single cell transcriptomic analyses were produced from fore- and hindlimbs. **B** UMAP clustering of wildtype and mutant fore- and hindlimbs shows one mesenchymal as well as four satellite clusters. **C** UMAP colored according to *Pitx1* expression in wildtype hindlimbs (levels represented by the red color scale) shows expression mostly in the mesenchyme cluster. **D** A cassette containing a minimal β-globin promoter (mP) and an EGFP reporter gene is integrated upstream of *Pitx1*. A secondary round of CRISPR/Cas9 targeting is then used to delete the *Pen* enhancer. **E** Conventional and light-sheet microscopy reveal that $Pitx1^{GFP}$ embryos display EGFP expression domains corresponding to the one of *Pitx1* ($N = 3$), scale bars = 2 mm. **F** RNA-seq and H3K27ac tracks of sorted hindlimb cells show that the sensor approach can separate *Pitx1* active (GFP+) and inactive (GFP−) regulatory landscapes. **G** C-HiC of the *Pitx1* locus in GFP+ and GFP− hindlimb cells. Darker red or blue bins represent more frequent contacts as represented by scaled bars on the left. GFP+ cells bear chromatin interactions between *Pitx1* and its associated enhancers (see green arrows). GFP− cells do not display these interactions but a strong contact between *Pitx1* and *Neurog1* (see red arrow). The lower map is a subtraction of the two above where GFP+ preferential interactions are displayed in red and GFP- preferential interactions in blue.

these cell types does not preclude a fraction of GFP−/*Pitx1*− to be of mesenchymal origin as we found a weak but clear expression of some mesenchymal markers such as *Prrx1* or *Twist1* in this population (Fig. S3C). Conversely, we found weak expression of muscle (*Myh3*) and ectodermal (*Krt14*) markers in GFP+/*Pitx1*+ cells (Fig. S3C). Finally, as *Pitx1* was previously associated with tissue outgrowth, we also assayed proliferative and apoptotic behaviors of GFP+ and GFP− cells[20]. As suspected, we found that GFP+/*Pitx1*+ cells are slightly more proliferative and contain less apoptotic cells (Fig. S4A, B).

We then assayed the *cis*-regulatory activities in GFP−/*Pitx1*− and GFP+/*Pitx1*+ hindlimb cells using the H3K27ac chromatin mark as a proxy for enhancer activities and C-HiC to determine the locus chromatin architecture[21]. In GFP−/*Pitx1*− cells, neither *Pitx1* promoter nor its various enhancers, including *Pen*, were

found enriched with H3K27ac (Fig. 1F). Moreover, the locus 3D structure is in a repressed state where *Pitx1* displays a strong interaction with the repressed *Neurog1* gene and no interaction with its cognate enhancers (Figs. 1G and S5). This data shows that GFP−/*Pitx1*− hindlimb cells display a complete absence of active regulatory landscape features. In contrast, in GFP+/*Pitx1*+ cells all known *Pitx1* enhancers as well as its promoter are strongly enriched in H3K27ac chromatin marks. Furthermore, in these cells *Pitx1* establishes strong contacts with its enhancers *PelB*, *PDE*, *RA4*, and *Pen* (Figs. 1F, G and S5).

In summary, this data shows that within the hindlimb, classically considered as a *Pitx1* active tissue, 8% of cells, from mesenchymal, immune, endothelium, muscle and epithelium origin, display an inactive *Pitx1* cis-regulatory landscape and 3D architecture. Moreover, it suggests a bimodal regulatory behavior,

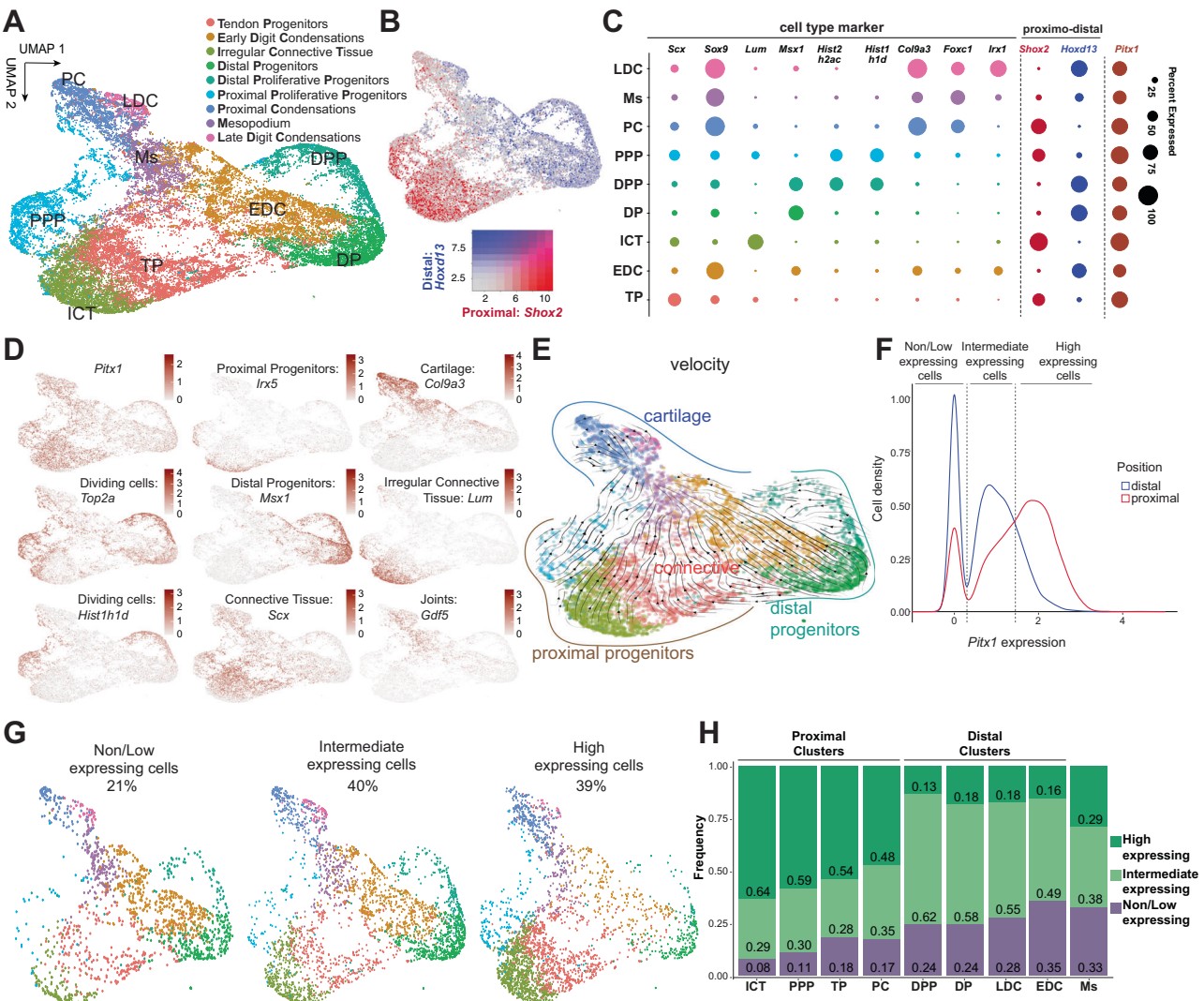

**Fig. 2 *Pitx1* expression in wildtype hindlimbs. A** UMAP of re-clustered mesenchymal cells from all datasets. **B** UMAP distribution of *Shox2* (proximal) and *Hoxd13* (distal) markers. Red to blue heatmap color scale represents levels of expression of *Shox2* or *Hoxd13*, respectively. **C** Representative marker genes for each cluster. The dot size corresponds to the percentage of cells that express a given marker in the hindlimb *Pitx1*^{+/+} dataset. **D** UMAP expression distribution of selected marker genes. Red color scales represent selected marker genes levels of expression. **E** RNA-velocity analysis of hindlimb wildtype mesenchymal clusters. Note that the differentiated chondrocytic cell clusters (upper part) are predicted to derive from proximal and distal progenitors' clusters (bottom part). **F** *Pitx1* expression density plot in the proximal (red line) and distal clusters (blue line) in the hindlimb wildtype dataset. Definition of the three types of *Pitx1*-expressing cells: non/low− (<= 0.3 *Pitx1* expression levels), intermediate- (>0.3; <=1.45), high- expressing (>1.45). The expression scale corresponds to a logE(expression_value+1). **G** Hindlimb wildtype cells distribution across clusters in the UMAP space based on *Pitx1* expression levels. **H** Hindlimb wildtype cells proportions according to *Pitx1* expression levels across mesenchymal clusters.

where the *Pitx1* promoter, its associated enhancers and the locus 3D structure are all displaying an *active* mode or none of them are. We then further characterised *Pitx1* expression specificities within the hindlimb mesenchyme.

**Hindlimb proximal cell clusters express *Pitx1* at higher level.** To characterize *Pitx1* transcription within mesenchymal subpopulations, we first re-clustered mesenchymal cells from all datasets. From this analysis, we could define nine clusters (Fig. 2A). We first observed that their distribution in the UMAP space is strongly influenced by the limb proximo-distal axis, as illustrated by *Shox2* (proximal marker) and *Hoxd13* (distal marker) transcript distributions (Fig. 2B). We further annotated the clusters according to the expression of known marker genes (Supplementary Dataset S1). In the proximal limb section, we identified four clusters. First, we found an undifferentiated **P**roximal **P**roliferative

**P**rogenitors cluster which is characterized by high expression of proliferative marker genes, where most cells were found in G2 and S phase and that expresses markers linked to previously identified limb mesenchymal progenitor (LMPs) cells (**PPP:** *Irx5+, Alx4+, Tbx2/3+, Shox2+, Hist1h1d+, Top2a+*) (Figs. 2C, D and S6A, B)[22]. We then identified a **T**endon **P**rogenitor cluster (**TP:** *Shox2+; Osr1+; Scx+*) and an **I**rregular **C**onnective **T**issue cluster which includes muscle connective tissue and ultimately patterns tendons and muscles (**ICT:** *Shox2+; Osr1+, Dcn+, Lum+, Kera+, Col3a1+*) (Fig. 2C, D)[23]. Finally, in the proximal limb we observed a single cluster of **P**roximal **C**ondensations, which already displays late chondrogenic markers and will give rise to proximal limb bones (**PC:** *Tbx15+; Sox9+; Col2a1+, Col9a3+, Acan+*) (Figs. 2C,D and S6B)[22]. In the distal limb, we observed the presence of two undifferentiated distal mesenchyme (*Msx1+*) clusters that also relate to previously identified LMPs: one that we classified as **D**istal **P**roliferative **P**rogenitors (**DPP:** *Tbx2/3+, Jag1+,*

*Hoxd13*+; *Msx1*+; *Hist1h1d*+) as it displays a strong expression of proliferation markers, while the other is defined as **D**istal **P**rogenitors (**DP**: *Tbx2/3*+, *Jag1*+, *Hoxd13*+; *Msx1*+) (Figs. 2C, D and S6B). In both of these clusters a majority of the cells appear to be either in G2 or S phase indicative of their high proliferative rate (Fig. S6A). Also, in the distal limb, we identified two more differentiated clusters: **E**arly **D**igit **C**ondensations (**EDC**: *Hoxd13*+; *Sox9*+, *Col2a1*−, *Col9a3*−), which are a type of distal osteo-chondrogenic progenitors and **L**ate **D**igit **C**ondensations (**LDC**: *Irx1*+, *Col2a1*+, *Col9a3*+) which are more differentiated chondrocytes[22]. Finally, in-between proximal and distal regions (*Shox2*+ and *Hoxd13*+), we found a cluster of chondrocytic cells that we considered to be the **Me**sopodium (**Ms**: *Sox9*+; *FoxcI*+, *Gdf5*+, *Col2a1*+) and thus corresponding to ankles or wrists (Fig. 2C, D, Supplementary Dataset S1).

To better understand the links between the different clusters, we ran an RNA velocity analysis in the hindlimb dataset, which predicts cell lineage differentiation based on the dynamics of spliced (mature) versus unspliced (immature) mRNAs (Fig. 2E)[24,25]. We found that in the proximal limb a set of *Irx5*-expressing cells located within the PPP and ICT clusters are progenitors for the more differentiated proximal clusters such as TP and PC (Fig. 2D, E)[26]. In the distal limb, DP and DPP clusters appear to be progenitors for EDC and then LDC. The Ms cluster originates from both proximal (PPP-ICT) and distal (DP-DPP) progenitor clusters, confirming its proximo-distal origin (Fig. 2E).

We then assessed whether *Pitx1* is differentially expressed among clusters in *Pitx1*$^{+/+}$ hindlimbs. Overall, we found *Pitx1* expressed in all mesenchymal clusters, yet with a proximal preference (Fig. 2D, F). We then classified mesenchymal *Pitx1* expressing cells in three categories: non/low-expressing (transcription values < =0.3, 21% of the hindlimb wildtype cells), intermediate-expressing (transcription > 0.3; <= 1.45, 40% of wildtype cells), and high-expressing (transcription > 1.45, 39% of wildtype cells) (Fig. 2F). As expected, we found that a majority of high expressing cells are located in proximal clusters (PPP, TP, ICT, PC) and a majority of intermediate-expressing cells in distal clusters (DP, DPP, EDC, LDC) (Fig. 2G, H). We also observed that the Ms cluster, previously identified as a cluster originating from the proximal and distal cell-types, is formed by a similar distribution of high-expressing (proximal) and intermediate-expressing (distal) cells in line with a proximo-distal origin (Fig. 2H).

***Pitx1* expression levels associate with global change in regulatory landscape acetylation.** Next we explored how cells can achieve distinct *Pitx1* transcriptional outputs. Practically, we asked whether high- and intermediate-expressing cells use a distinct *Pitx1* enhancer repertoire to account for the different expression levels. We sorted the two cell populations from *Pitx1*$^{GFP}$ hindlimbs by GFP intensities: GFP+− (intermediate-expressing) and GFP++ (high-expressing) and performed RNA-seq as well as H3K27ac ChIP-seq on the two positive populations (Fig. 3A). On average, we found three times more *Pitx1* transcripts in GFP++ cells than in GFP+− cells as well as an enrichment for several known *Pitx1* target genes including *Tbx4* (Fig. 3B, Fig. S7A,B, Supplementary Dataset S4)[8]. Moreover, as expected from the single-cell analysis, high-expressing GFP++ cells were mostly enriched for proximal limbs markers (*Shox2*, *Gsc*, *Tbx18*, *Tbx3*, and *Hoxa11*) and showed higher expression of ICT marker genes (*Kera* and *Lum*) (Figs. 3C, and S7C, Supplementary Dataset S4). In contrast, intermediate-expressing cells GFP+− where enriched for distal cell markers (*Hoxa13*, *Hoxd13*, *Wnt5a*, *Lhx2* and *Msx1*) (Figs. 3C, S7C, Supplementary Dataset S4).

In both intermediate- and high-expressing cells, the previously characterized *Pitx1* enhancer repertoire—*PelB*, *PDE*, *RA4*, and *Pen*—was found marked by H3K27ac. Yet, in high-expressing cells (GFP++), stronger H3K27ac signal was found at these elements concomitantly with a strong increase at two specific regions: the *Pitx1* proximal promoter region (*PPPR*) and the region A (*regA*). We also observed a few regions upstream of *Pen* that were strongly enriched for H3K27ac in GFP++ cells (Fig. 3D, E)[11]. However, those sequences do not seem to be important for *Pitx1* expression as the deletion of the entire region between *Pitx1* and *Pen*, including *Pen* but not those regions, fully recapitulates the *Pitx1* hindlimb knock out phenotype[11]. Altogether, this data shows that *Pitx1* regional expression differences across hindlimbs associate with a progressive increase of its *cis*-regulatory landscape activity rather than from the usage of different enhancers repertoires. These results further re-enforce the idea that the fundamental unit of *Pitx1* regulation is the landscape as a whole rather than individual enhancers.

***Pen* deletion increases the proportion of *Pitx1* non/low-expressing cells in hindlimbs.** Observing the coordination between regulatory units at the locus to modulate gene expression, we sought to test how the deletion of one of them influences the overall unity of the locus. Therefore, we took advantage of the *Pitx1* EGFP sensor and of scRNA-seq to track how the homozygous deletion of the *Pen* enhancer affects the hindlimb *Pitx1* locus activity. First, the removal of *Pen* within the *Pitx1*$^{GFP}$ background (*Pitx1*$^{GFP;ΔPen}$) induced a shift in the expression of the GFP reporter gene in hindlimbs (Fig. 4A, B). Specifically, the proportion of GFP− cells raised from 8% in *Pitx1*$^{GFP}$ to 16% in *Pitx1*$^{GFP;ΔPen}$ at E12.5 and from 12% to 29% at E13.5 (Fig. S8A, B). To confirm that this effect is not due to a difference in the distribution of EGFP fluorescence during cell sorting, we compared EGFP transcription in *Pitx1*$^{GFP;ΔPen}$ and *Pitx1*$^{GFP}$ GFP− cells and did not observe a difference (Fig. S8C).

Secondly, we compared *Pitx1*$^{+/+}$ and *Pitx1*$^{Pen−/Pen−}$ scRNA-seq dataset and found a similar effect as the *Pen* deletion induces a significant 29% loss of *Pitx1* expression (adjusted *p*-value = 1.75e−96 (Wilcoxon Rank Sum test)) featured by a decrease in *Pitx1* high-expressing cells and a strong increase in non/low-expressing cells (Fig. 4C). Across hindlimb mesenchymal cells, the proportion of non/low-expressing cells was indeed raised from 21% in *Pitx1*$^{+/+}$ to 35% in *Pitx1*$^{Pen−/Pen−}$. In summary, the two approaches show that behind the weak average loss of *Pitx1* expression, a strong increase of non/low-expressing cells in mutant hindlimbs could account for the clubfoot phenotype seen in these animals[11].

We further quantified within the scRNA-seq dataset if this alteration in expression was equally distributed among various hindlimb cell-types or if some populations were more specifically affected. All clusters with the exception of the Ms and the LDC showed a significant loss of *Pitx1* expression ranging from 24 to 39% (Fig. 4D). With respect to the proportion of non/low-expressing cells, we saw that proximal cells showed a preferential 2.1-fold enrichment of non/low-expressing cells (13 to 28%) in comparison with distal cells (1.6-fold, 29 to 45%) (Fig. 4E, F). We then computed the increase of non/low-expressing *Pitx1* cells in each cluster and saw that two proximal clusters, ICT and PPP, showed a particularly strong 3.5- and 2-fold increase in *Pitx1* non/low-expressing cells respectively (Fig. 4D). It is important to note that in both clusters the vast majority of cells usually express *Pitx1* at a high level (Figs. 2H, S9). Other clusters showed 1.5- to 1.8-fold increase in *Pitx1* non/low-expressing cells. In conclusion, we found that proximal, high-expressing clusters are more affected by the enhancer deletion than distal,

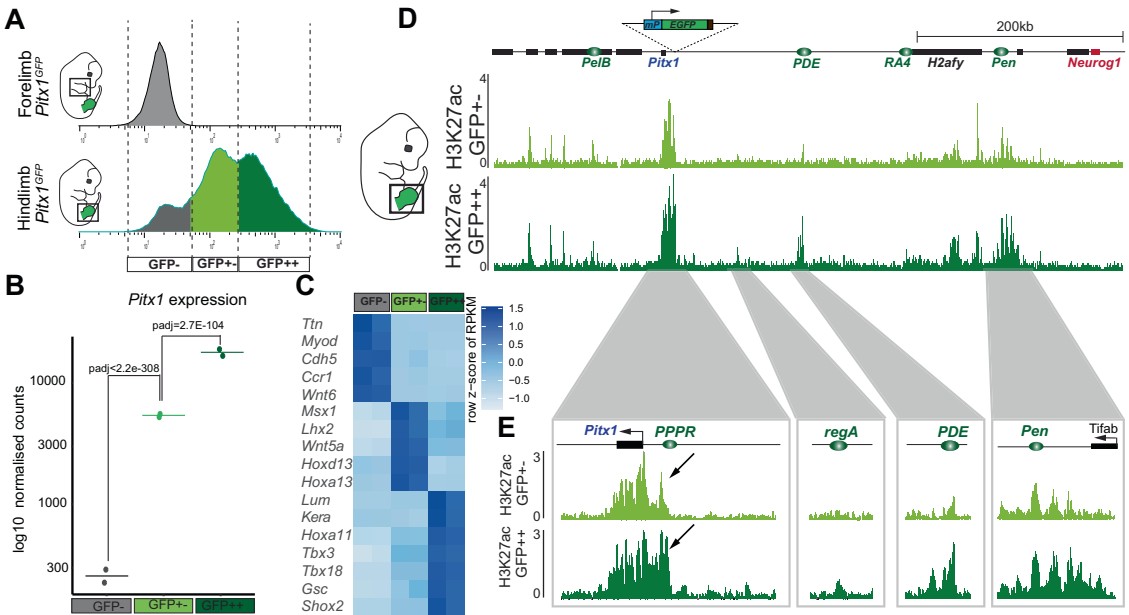

**Fig. 3 High- and intermediate-expressing *Pitx1* regulatory landscape activities. A** FACS sorting of *Pitx1^{GFPs/GFPs}* forelimb and hindlimbs. Note the EGFP high- (GFP++, dark green) and intermediate-expressing (GFP+−, light green) populations. **B**. Normalised counts of *Pitx1* expression in GFP−, GFP+− and GFP++ cells. Averages are represented by a horizontal bar. Adjusted *p*-values (padj) of differential gene expression are computed using the Wald-test and Benjamini-Hochberg multiple testing correction as implemented in the Deseq2 tool (Supplementary Dataset S4 and Source data). **C** Expression of selected marker genes in GFP− (satellite cell types), GFP+− (distal) and GFP++ (proximal and ICT) cells. **D** H3K27ac ChIP-seq at the *Pitx1* locus. Note that enhancers are active in both intermediate- and high-expressing cells, yet with a few regions marked only in high-expressing cells. **E** H3K27ac profile at the *Pitx1* gene body, *Pitx1* Proximal Promoter Region (PPPR, see black arrow), region A (regA), *PDE* and *Pen* enhancer.

intermediate-expressing clusters. We subsequently investigated if this differential alteration of *Pitx1* expression among hindlimb cell population affected the proportion of cells within the clusters.

**Pen deletion delays the formation of irregular connective tissue**. As a positive control for the effect of *Pitx1* loss-of-function in *Pitx1^{Pen−/Pen−}* embryos, we took advantage of two datasets that do not express *Pitx1*: *Pitx1^{+/+}* forelimbs and *Pitx1^{−/−}* hindlimbs. As a proxy for the functional impact of *Pitx1* transcriptional change on limb development, we measured the relative proportions of the different cell clusters in the different datasets. First, we did not observe changes in the proportions of non-mesenchymal satellite cell clusters in any of the conditions (Fig. S10). We then measured the proportions of the different mesenchymal sub-clusters (Fig. 5A, B). By comparing wildtype fore- and hindlimbs, we did not observe any significant change in the proportion of cell-types, suggesting that fore- and hindlimbs are similarly populated despite the obvious structural differences between arms and legs. In contrast, *Pitx1^{−/−}* hindlimbs display a heterochronic phenotype, featuring an increase in progenitor cells in both the proximal and distal regions of hindlimbs (PPP and DPP cell clusters) while a concurrent decrease is seen in several differentiated cell types in proximal and distal hindlimbs (ICT, PC, Ms, and LDC) (Fig. 5A, B). Remarkably, the loss of the *Pen* enhancer resulted in a similar effect but only significant in the proximal limb cell clusters (Fig. 5A, B). Specifically, the proportion of PPP cells increased in *Pitx1^{Pen−/Pen−}* hindlimbs as the proportion of ICT cells decreased. This alteration correlates with the strong loss of *Pitx1* transcription seen in both clusters (Fig. 4D). The increase in PPP cells is further supported by the upregulation of its markers in *Pitx1^{Pen−/Pen−}* hindlimbs (*Hist1h* genes, *Top2a*, and others; Supplementary Dataset S5).

To test if these effects could be explained by a delayed differentiation of progenitor cells, we performed a velocity

analysis on *Pitx1^{−/−}*, *Pitx1^{Pen−/Pen−}* and *Pitx1^{+/+}* limbs in proximal and distal cell clusters separately. In the proximal part of the hindlimb, we found in both mutants a predicted connection from PPP to ICT cells, suggesting an ongoing differentiation process (Fig. 5C). This connection was not present in *Pitx1^{+/+}* fore- and hindlimbs, suggesting that the differentiation process was completed in these tissues. These findings are further supported by an increase of ICT marker genes (*Lum*, *Dcn*, and *Kera*) in PPP cells of both mutants, suggesting that those cells have only partially adopted an ICT identity but still did not fully differentiate (Fig. S11). In contrast, the velocity analysis of distal clusters did not show any changes in *Pitx1^{Pen−/Pen−}* hindlimbs, in agreement with the unaltered proportion of distal cell clusters shown above (Fig. 5B, D). Finally, we observed in *Pitx1^{−/−}* hindlimbs an accumulation of distal progenitor cells and a loss of differentiated LDC cells suggesting a slower distal differentiation process in *Pitx1^{−/−}* hindlimbs. Together, these findings support a form of heterochrony that affects only the proximal part of *Pitx1^{Pen−/Pen−}* hindlimbs and that is featured by a delayed differentiation of PPP to ICT.

As *Pitx1* has been shown to have both indirect and direct downstream effects, we further investigated differentially expressed genes in *Pitx1* loss-of-function hindlimbs that could induce these effects. In particular it has been shown that *Tbx4*, a known downstream target gene of *Pitx1*, mediates the *Pitx1*-effect on hindlimb buds growth rate[8,20,27]. As anticipated, we found a downregulation of the *Tbx4* in all clusters aside of PC, Ms, and LDC in both *Pitx1^{−/−}* and *Pitx1^{Pen−/Pen−}* hindlimbs, with the strongest effect in ICT and PPP clusters (Fig. S12A–D, Supplementary Dataset S5). To further determine the origin of the *Tbx4* loss we assessed in *Pitx1^{Pen−/Pen−}* hindlimb clusters the expression of caudal *Hox* genes, which have been suggested to control *Tbx4* along with *Pitx1*[28]. Here, we did not find an alteration in *Hox* expression levels that correlates with *Tbx4* loss, suggesting that *Tbx4* decrease is rather a direct effect of *Pitx1*

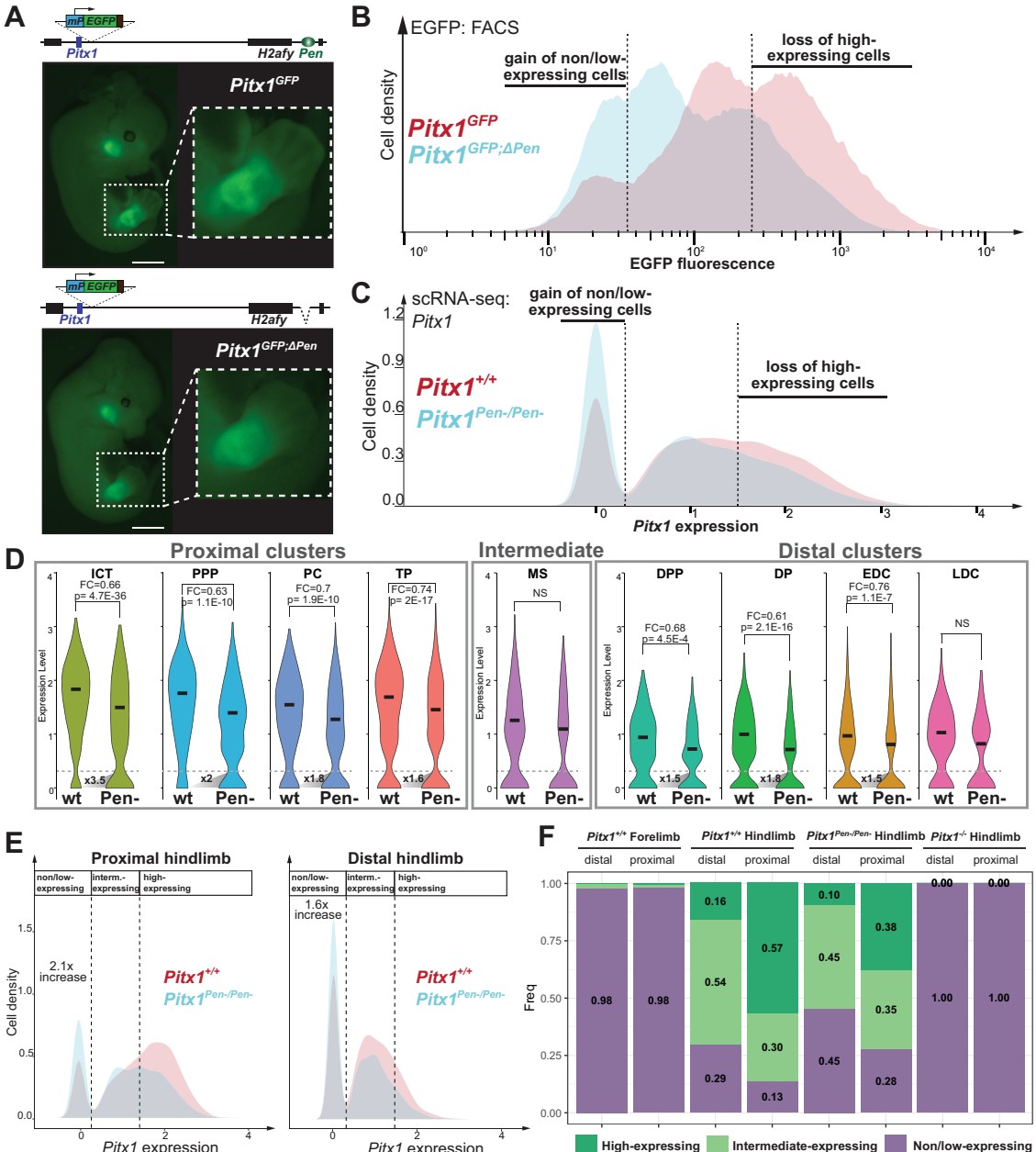

**Fig. 4 Influence of the *Pen* deletion on *Pitx1* expression in hindlimb cell populations. A** EGFP expression pattern in *Pitx1^GFP^* and *Pitx1^GFP;ΔPen^* E12.5 embryos (*N* = 3), scale bar = 2 mm. **B** FACS profile of *Pitx1^GFP^* (red) and *Pitx1^GFP;ΔPen^* (cyan) hindlimbs shows an increased number of EGFP non/low-expressing cells as well as a decrease of EGFP high-expressing cells in *Pitx1^GFP;ΔPen^* hindlimbs. **C** *Pitx1* expression distributions in *Pitx1^+/+^* (red) and *Pitx1^Pen−/Pen−^*(cyan) hindlimb cells show an increased proportion of non/low-expressing cells and a decreased proportion of high-expressing cells in *Pitx1^Pen−/Pen−^* hindlimbs. The dotted lines indicate the threshold between non/low-expressing and intermediate-expressing (0.3) as well as between intermediate-expressing and high expressing cells (1.45). The expression scale is a logE(expression_value+1). **D** *Pitx1* expression across all clusters in *Pitx1^+/+^* and *Pitx1^Pen−/Pen−^* hindlimbs. Averages are represented by a horizontal bar. Adjusted *p*-values (*p*) shown in the figure were calculated by Wilcoxon Ranks Sum test using the FindMarkers function from Seurat R Package (Supplementary Dataset S5). The dotted lines indicate the threshold between non/low-expressing and intermediate-expressing (0.3). The fold change in non/low-expressing cell number between *Pitx1^+/+^* and *Pitx1^Pen−/Pen−^* is shown at the bottom of the violin plots. Note the strong loss of expression and the accumulation of non/low-expressing cells in ICT and PPP clusters. **E** Distribution of *Pitx1* expression in proximal and distal cells of *Pitx1^+/+^* and *Pitx1^Pen−/Pen−^* hindlimbs. Note the strong increase in proximal non/low-expressing cell fraction. **F** Proportion of non/low-, intermediate- and high-*Pitx1* expressing cells across conditions.

loss-of-expression (Supplementary Dataset S5). Finally, we measured if the *Tbx4* expression loss was sufficient to alter cell proliferation and apoptosis, therefore changing the hindlimb cell type composition[20]. Overall, we did not observe changes in either, suggesting that the observed loss of *Tbx4* is not sufficient to alter cell proliferation and apoptosis, and that the induced phenotype takes origin from an independent mechanism (Fig. S13A, B).

Moreover, aside from *Tbx4*, we found numerous dysregulated genes in *Pitx1^Pen−/Pen−^* hindlimbs which might contribute to observed phenotypes (Supplementary Dataset S5). This is the case for *Dcn*, an ICT marker gene previously described to be involved in tendon elasticity in mice as well as the *Six1* and *Six2* genes that are expressed in connective tissue and necessary for skeletal muscle development[23,29–32].

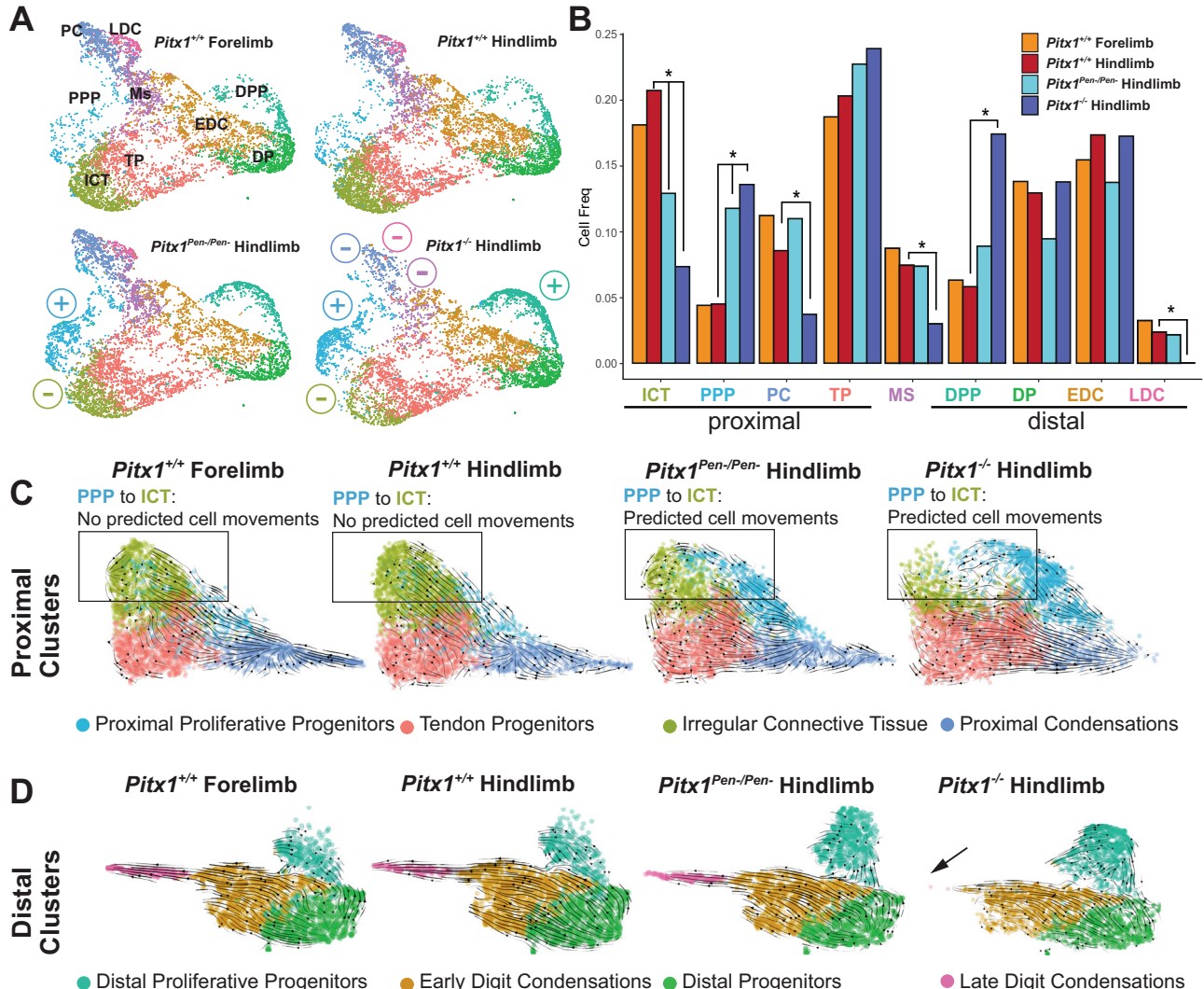

**Fig. 5 Influence of the _Pen_ deletion on limb cell populations. A** UMAP of mesenchymal cell type proportions across conditions, (+) and (−) symbols indicate increase or decrease in cell proportions in comparison to wildtype hindlimbs. Abbreviations are described in Fig. 2A. **B** Quantification of cell type proportions across conditions, *p*-values < 0.01 are marked with an asterisk (see Source data). *P*-values were calculated pairwise using differential proportion analysis in R[60]. Abbreviations are described in Fig. 2A. **C** Velocity analyses from proximal clusters in all dataset. Arrows represent the direction toward the predicted fate territories. Note in *Pitx1^{Pen−/Pen−}* and *Pitx1^{−/−}* the velocity-predicted cell movements between PPP and ICT which likely represent ongoing differentiation that is not predicted in wildtype hindlimbs or forelimbs. **D** Velocity analyses from distal clusters in all dataset. Note in *Pitx1^{−/−}* hindlimbs the loss of late digit condensations (black arrows) at the end of the velocity-predicted cell movements.

**The _Pen_ enhancer contributes to _Pitx1_ regulatory landscape activation.** The establishment of the active *Pitx1* chromatin landscape includes changes in 3D conformation and the acetylation of specific *cis*-regulatory elements. Therefore, we asked whether the *Pen* enhancer itself is required to establish these features and specifically if its deletion would impact them.

In GFP+ and GFP− cells from *Pitx1^{GFP;ΔPen}* hindlimb buds, we used RNA-seq to assess whether we could observe similar changes in cellular identity upon *Pen* enhancer loss as the one previously described using scRNA-seq. As expected, we could observe in GFP− cells the accumulation of mesenchymal markers (*Prrx1*, *Twist1*) with a particular enrichment for ICT markers (*Col3a1, Col1a1, Col1a2, Lum*) (Fig. 6A, Supplementary Dataset S6). As a consequence of the accumulation of *Pitx1* non/low-expressing mesenchymal cells, we also observed a dilution of non-mesenchymal clusters marked by a decrease of epithelium (*Wnt6, Krt14*) and muscle (*Ttn*) markers. In GFP+ cells, we did not observe a clear change in identity markers indicating that the cell type composition is similar between

*Pitx1^{GFP}* and *Pitx1^{GFP;ΔPen}* high-expressing cells (Supplementary Dataset S7). This suggests that these high expressing cells, that *escape* a loss-of-expression following the deletion of *Pen*, must display an adaptive mechanism to accommodate the *Pen* enhancer loss.

We then performed H3K27ac ChIP-seq in the escaping GFP+ cells and in the increased fraction of GFP− cells. In *Pitx1^{GFP;ΔPen}* GFP+ cells, we observed a distribution of H3K27ac over the landscape that was virtually identical to *Pitx1^{GFP}* GFP+ hindlimbs cells, with the exception of the *Pen* enhancer itself (Fig. 6B). This result suggests that the *Pitx1* expressing cells in the *Pen* deletion background use the same enhancer repertoire as the *Pitx1^{GFP}* expressing cells and thus do not use an alternative regulatory landscape. Moreover, we observed the same average *Pitx1* expression level in *Pitx1^{GFP}* and *Pitx1^{GFP;ΔPen}* GFP+ cells (Supplementary Dataset S7). In GFP− cells deleted for *Pen*, in contrast to *Pitx1^{GFP}* cells, we observed ectopic acetylation of the *Pitx1* promoter as well as of the *RA4* and *PelB* enhancers (Fig. 6C). These activities are likely caused by the relocation, in

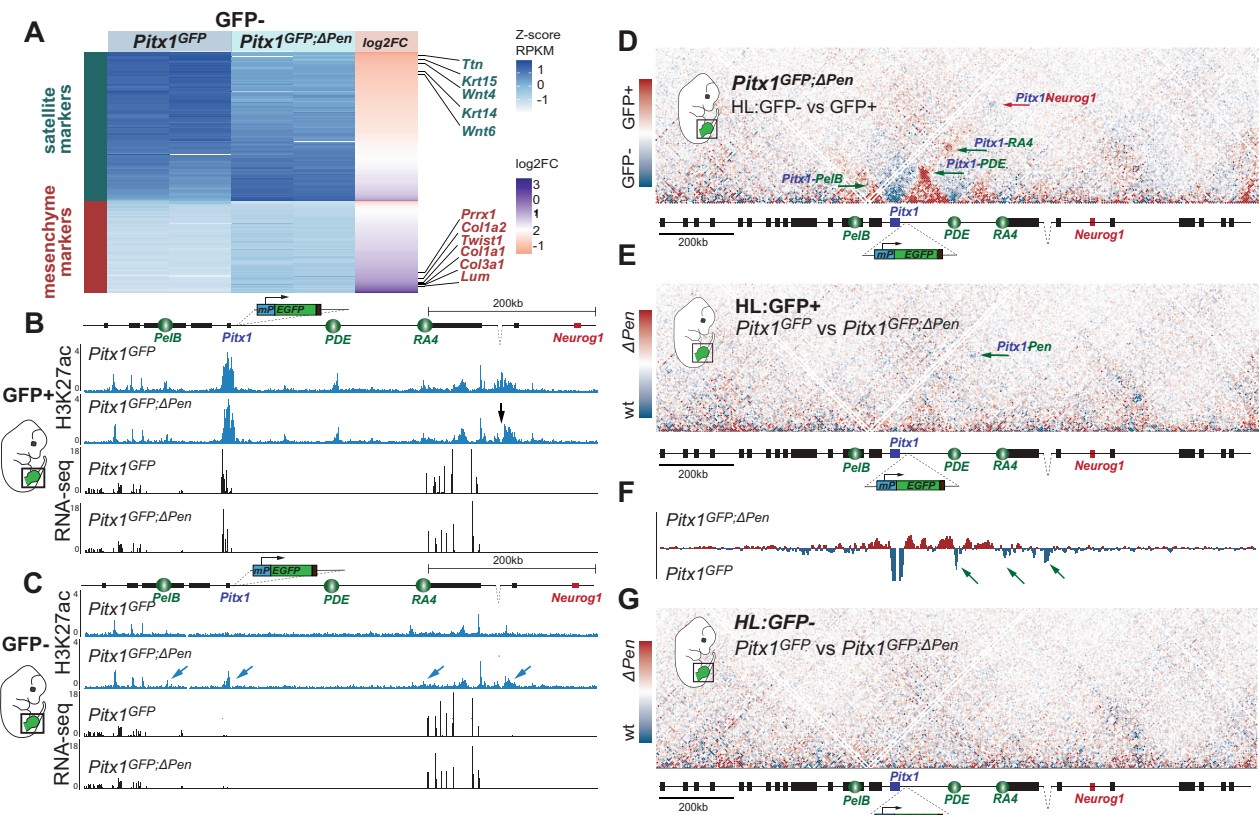

**Fig. 6 Single enhancer deletion results in inefficient regulatory landscape activation. A** Log2 fold change and RPKM of mesenchymal (red) and satellite (darkgreen) marker genes in *Pitx1^GFP^* and *Pitx1^GFP;ΔPen^* GFP− hindlimbs cells. Note the decrease in satellite markers and the increase in mesenchymal markers in *Pitx1^GFP;ΔPen^* GFP− cells. **B** H3K27ac ChIP-seq and RNA-seq tracks at the *Pitx1* locus in GFP+ cells of *Pitx1^GFP^* and *Pitx1^GFP;ΔPen^* hindlimbs. Note the loss of the *Pen* enhancer region (black arrow). **C** H3K27ac ChIP-seq and RNA-seq at the *Pitx1* locus in GFP− cells of *Pitx1^GFP^* and *Pitx1^GFP;ΔPen^* hindlimbs. Note the acetylation of *Pitx1* promoter and enhancers (blue arrows) and the weak *Pitx1* transcription. **D** C-HiC subtraction map between GFP− and GFP+ *Pitx1^GFP;ΔPen^* hindlimb cells. GFP+ preferential interactions are displayed in red and GFP- preferential interactions in blue. **E** C-HiC subtraction maps between *Pitx1^GFP^* and *Pitx1^GFP;ΔPen^* GFP+ hindlimb cells. Note the loss of interaction between *Pitx1* and the *Pen*-deleted region (green arrow). *Pitx1^GFP;ΔPen^* GFP+ preferential interactions are displayed in red and *Pitx1^GFP^* GFP+ preferential interactions in blue. **F** Subtraction track of virtual 4C between *Pitx1^GFP^* (blue) and *Pitx1^GFP;ΔPen^* (red) GFP+ hindlimb cells with the *Pitx1* promoter as viewpoint. Note the partial loss of interactions between *Pitx1* and its telomeric enhancers (*PDE*, *RA4*, and *Pen*; green arrows). **G** C-HiC subtraction maps between *Pitx1^GFP^* and *Pitx1^GFP;ΔPen^* GFP− hindlimb cells. *Pitx1^GFP;ΔPen^* GFP− preferential interactions are displayed in red and *Pitx1^GFP^* GFP− preferential interactions in blue.

the GFP− fraction, of cells that would normally express *Pitx1* but fail to establish a fully active landscape in the absence of *Pen*. In these cells, we observed a marginal increase in *Pitx1* expression (FC = 1.6, padj = 0.0026) that suggests that the locus is less repressed as in wildtype GFP− cells (Fig. 6C, Supplementary Dataset S6).

We then measured how the lack of *Pen* affects the locus 3D structure dynamics in *Pitx1^GFP;ΔPen^* hindlimbs. First, GFP+ and GFP− *Pitx1^GFP;ΔPen^* hindlimb cells displayed differences similar to their *Pitx1^GFP^* active and inactive counterparts (Figs. 1F, 6D, S14A). This suggests that *escaping* high-expressing hindlimb *Pitx1^GFP;ΔPen^* cells do not require *Pen* to establish an active 3D conformation. We then asked whether these cells bare an alternative chromatin structure than wildtype ones to compensate for the loss of *Pen*. By comparing *Pitx1^GFP^* and *Pitx1^GFP;ΔPen^* GFP + cells we saw no major differences (Figs. 6E, S14B). Yet, using virtual 4C, we saw a slight reduction of contacts between the *Pitx1* promoter and *PDE/RA4* in GFP+ cells (Figs. 6F, S14D). This suggests that the remaining high-expressing cells do not necessarily undergo a strong adaptive structural response to the loss of *Pen* to ensure high *Pitx1* expression. Finally, we asked whether the relocated *Pitx1^GFP;ΔPen^* GFP− cells, that bear ectopic promoter and enhancer acetylation, display features of an active 3D structure (Figs. 6G, S14C, D). However, we did not observe

any changes in the *Pitx1* locus conformation in these cells in comparison to *Pitx1^GFP^* GFP− cells. This shows that despite some remaining regulatory activity (evidenced by low level H3K27ac; arrows Fig. 6C), the locus is unable to assume its active 3D structure and therefore to efficiently transcribe *Pitx1* (Fig. 6G). In conclusion, the *Pen* enhancer is necessary to ensure that all the cells with active enhancers at the *Pitx1* locus undergo a robust transition toward a structurally and transcriptionally active landscape (Fig. 7).

## Discussion

In this work we have shown that hindlimb cells display several states of *Pitx1* regulatory activities. In active cells, all enhancers are marked with the active H3K27ac chromatin modification and are contacting the *Pitx1* promoter. In contrast, in inactive cells, we could not observe partial regulatory activities, i.e. neither enhancer acetylation nor enhancer-promoter interactions. This shows that the locus follows a bimodal behavior where the regulatory landscape as a whole acts on *Pitx1* transcription. Indeed, a common set of coordinated enhancers are active in both proximal *Pitx1* high-expressing and distal *Pitx1* low-expressing cells. In fact, the *Pitx1* regulatory landscape acts here similarly to what was previously defined as a holo-enhancer, where the whole

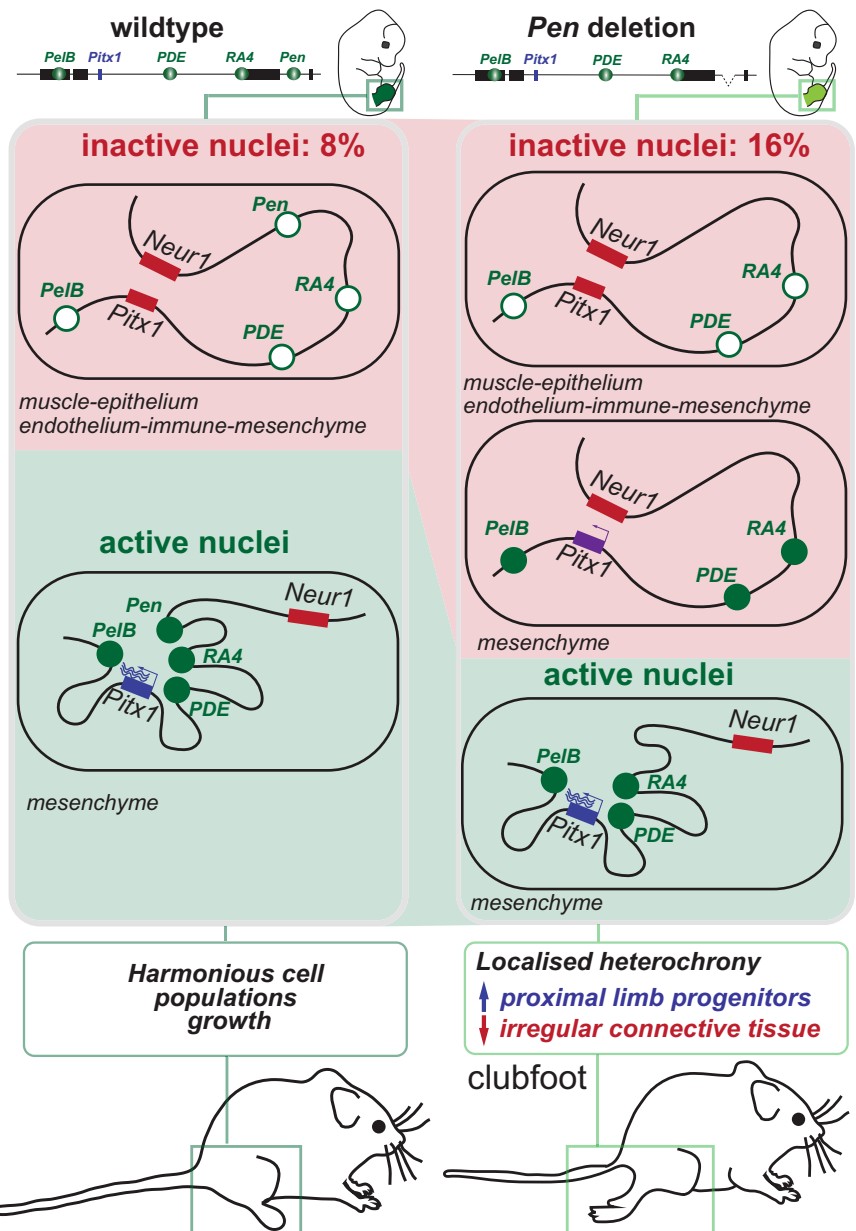

**Fig. 7 Model.** In wildtype hindlimb tissues (left panel) 8% of the nuclei, mostly from non-mesenchymal origin, display a completely repressed *Pitx1* locus, featured by inactive enhancers (white ovals), polycomb-repressed *Pitx1* gene (red rectangle) and inactive 3D chromatin structure (*Pitx1* does not contact its enhancers but contacts the repressed *Neurog1* gene). In active nuclei, the situation is inverted with active enhancers (green ovals), active 3D chromatin structure (*Pitx1* contacts its enhancers) and strong *Pitx1* transcription. In contrast, in hindlimb lacking the *Pen* enhancer (right panel), 16% of the cells are lacking *Pitx1* transcription. Among these cells, some display a partially active regulatory landscape. These latter cells, that have failed to establish an active 3D structure and a strong *Pitx1* transcription, are of mesenchymal origins in particular of ICT and PPP types. The remaining active cells in mutant hindlimbs appear to display wildtype expression levels. Phenotypically, the effect of the enhancer deletion is a disharmonious outgrowth of cell populations featured by a gain of PPP and a decrease of ICT cells. This cellular phenotype is likely at the origin of the clubfoot phenotype.

region seems to work as a coherent regulatory ensemble[33]. In this perspective, *Pitx1* expression levels are adjusted by the entire landscape. This is what we observed in high *Pitx1*-expressing proximal cells where the same enhancer set than in distal cells displays a higher enrichment for the active H3K27ac chromatin mark along with a few proximal-specific regions that are more enriched for H3K27ac. This suggests that proximal transcription factors or signaling cues are controlling the landscape either by binding simultaneously at several *Pitx1* cis-regulatory regions or by targetedly modifying other parameters of locus activity such as the frequencies of active chromatin interactions or the proximity to the repressive nuclear lamina for instance.

Here we have tested how the loss of one of the regulatory elements, the *Pen* enhancer, which is conserved among all tetrapods and required for hindlimb identity, affects the establishment of the *Pitx1* active landscape[11,34]. Some escaping cells can induce *Pitx1* regulatory landscape activation without *Pen*, suggesting that the other *cis*-regulatory modules (*PelB*, *PDE*, and *RA4*) provide a form a compensation. These modules are likely activated by a similar gene-regulatory network in wildtype and mutant hindlimbs, as we could not observe a clear shift in cell identity of *GFP/Pitx1* expressing cells. Alternatively, a cumulative effect of marginal transcriptional changes in cell identity along with specific non-transcriptional identity differences could

maintain the capacity of cells to generate a high *Pitx1* expression level despite the absence of *Pen*. Simultaneously, many *Pitx1* low/non-expressing cells accumulate in hindlimbs that bear enrichment of H3K27ac at the *Pitx1* promoter and at several of its enhancers (Fig. 7). Despite the presence of this active modification, the *Pitx1* locus does not adopt an active 3D chromatin folding but maintains the hallmarks of its inactive configuration. In fact, these accumulated low/non-expressing cells are seemingly stuck in a limbo between activity and repression and show the importance of the coordinated action of enhancer activity and 3D chromatin changes to achieve sufficient transcriptional strength. Therefore, we hypothesize that the role of *Pen* is not to act as a pattern-defining enhancer but rather as a support enhancer that ensures a robust transition of cells towards a fully active *Pitx1* landscape and therefore a strong *Pitx1* transcription. Here, other enhancers, such as *RA4*, *PelB* as well as other to be defined enhancers, might bear this pattern-defining role. In fact, *Pen* is a good model to understand the fundamental role of many enhancers that were characterised with a diverging activity than the gene they control[35–37]. This "class" of enhancers would therefore govern the cooperativity of loci's regulatory landscapes without defining by themselves their expression specificities.

Changes in the number of cells that express *Pitx1* in the hindlimb have strong phenotypical consequences. In fact, the complete loss of *Pitx1* induces an increase in proximal and distal progenitor cells concomitantly with a loss of differentiated cell types, overall altering the proportion of specific cell clusters in hindlimbs. The global increase in progenitors indicates a heterochrony in limb development that ultimately results in a reduction of the limb size and the loss of some limb structures such as the patella. In the case of the *Pen* enhancer deletion, we saw an enrichment of *Pitx1* low/non-expressing cells in PPP and ICT clusters resulting in a delayed differentiation of PPP into ICT. Although the reduction of *Pitx1* transcription induces the decrease of its direct target, *Tbx4*, involved in limb bud outgrowth and cell proliferation, it was here not sufficient to alter cell proliferation or apoptotic rates, at E12.5, suggesting that the clubfoot phenotype builds on a *Tbx4*-independent differentiation problem[20,27]. Here, the particularly strong effect of the *Pen* deletion on the ICT cell proportion pinpoints these cells as the origin of the clubfoot phenotype seen in mice lacking the enhancer. In fact, ICT has been repeatedly reported to function in a non-cell autonomous way during limb development and to act as an important driver of muscle patterning[23,38–43]. We, therefore, suspect that the loss of ICT in hindlimbs leads to a muscle patterning defect which would be at the base of the clubfoot phenotype. Moreover, the observed heterochrony in several of the mesenchymal cell populations could collectively cause the clubfoot since coordinated expansion and interactions among different mesenchymal cell populations are required for normal limb morphogenesis. Finally, despite lacking *Pitx1* expression as well, forelimb cell clusters are present in the same proportion as hindlimb ones. This suggests that the role of *Pitx1* in hindlimbs is mirrored by other genes in forelimbs, such as *Tbx5*, that account for a harmonious outgrowth of the various cell populations. Indeed, *Tbx5* loss of expression in the ICT population alters muscle and tendons patterning causing the mice to hold the paw in a supine position, leading them to walk on the edge or dorsal surface of the paw, resembling a clubfoot phenotype[23].

Our characterization of a single enhancer loss-of-function mutant at a cell subpopulation level opens the way to study the effect of other regulatory mutations with the same resolution and, in particular, of gain-of-function mutations. Such approaches will enable to select particular cell-subpopulations that show ectopic transcription in comparison to neighboring cells that bear the same mutation but no ectopic expression. This will facilitate a precise definition of features that are permissive for transcriptional gain-of-function and will be an important tool to further investigate the relationship between 3D structure, chromatin modifications, and gene transcriptional activation.

## Methods

### Cell culture, mice and tissue processing

*Animal procedures.* All animal procedures were in accordance with institutional, state, and government regulations (Canton de Genève authorisation: GE/89/19).

*CRISPR/Cas9 engineered alleles.* Genetically engineered alleles were generated using the CRISPR/Cas9 editing according to ref. [44]. Briefly, sgRNAs were designed using the online software Benchling and were chosen based on predicted off-target and on-target scores. All sgRNAs and target genomic locations for CRISPR–Cas9 can be found in Supplementary Table S1. SgRNAs were then sub-cloned in the pX459 plasmid from Addgene and 8 µg of each vector was used for mESCs transfection. mESCs culture and genetic editing followed standard procedure[45]. To construct the *Pitx1*^GFP mESCs clone, the LacZ sensor from[11] was adapted by exchanging the LacZ by an EGFP cassette. The sgRNA was designed to target CRISPR–Cas9 to chr13:55935371-55935390 (Supplementary Table S1). Cells were transfected with 4 µg of EGFP−cassette and 8 µg of pX459 vector containing the sgRNA. Transgenic G4 ESCs clones can be obtained upon request.

*Aggregation of mESC.* Embryos were generated by tetraploid complementation from G4 male ESCs obtained from the Nagy laboratory (http://research.lunenfeld.ca/nagy/?page=mouse%20ES%20cells)[17,46]. Desired mESCs were thawed, seeded on male and female CD1 feeders and grown for 2 days before the aggregation procedure. Donor tetraploid embryos were provided from in vitro fertilisation using c57bl6J x B6D2F1 backgrounds. Aggregated embryos were transferred into CD1 foster females. All animals were obtained from Janvier laboratories.

*Single-cell RNA-seq dissociation.* Two replicates of fore and hindlimb buds of E12.5 wildtype embryos and hindlimb buds of mutant embryos (*Pitx1*^Pen−/Pen−, *Pitx1*^−/−) were micro-dissected and incubated for 12 min in 400 µl trypsin-EDTA 0.25% (Thermo Fischer Scientific, 25300062), supplemented with 40 µl of 5% BSA. During incubation tissues were disrupted by pipetting after 6 min of incubation and at the end of the 12 min. Trypsin was then inactivated by adding 2× volume of 5% BSA and single cell suspension was obtained by passing cells in a 40 µm cell strainer. Cells were then spun at $250 \times g$ for 5 min at 4 °C and resuspended in 1%BSA in PBS. Cells were then counted using an automatized cell counter and a 1% BSA 700 cells/µl suspension was prepared. 10 µl of this solution was used as input for the 10× Genomics library preparation.

*Single-cell library preparation.* Single-cell libraries were prepared using the Chromium Single Cell 3′ GEM, Library & Gel Bead Kit v3 following the manufacture's protocol (10× Genomics, PN-1000075). Briefly, Gel beads in EMulsion (GEMs) are generated by combining Single Cell 3′ v3 Gel Beads, a Master Mix containing cells, and Partitioning Oil onto Chromium Chip B. Incubation of the GEMs produced from the poly-adenylated mRNA barcoded, full-length cDNA. Immediately, gel beads are dissolved and cDNA is amplified via PCR followed by library construction and sequencing. Libraries were paired-end sequenced on a HiSeq 4000. On average, 7000 cells were loaded on the Chromium Chip and between 25,000 and 35,000 mean reads were obtained.

*Whole-mount in situ hybridization (WISH).* Pitx1 WISH were performed on 40–45 somite stage mouse embryos (E12.5) using a digoxigenin-labeled *Pitx1* antisense riboprobe transcribed from a cloned *Pitx1* probe (PCR DIG Probe Synthesis Kit, Roche), as previously described in ref. [11].

*Light sheet microscopy imaging.* E12 embryos post-fixed overnight in 4% PFA. Tissue was cleared using passive CLARITY based clearing method. Briefly, tissue was incubated in a Bis-free Hydrogel X-CLARITY™ Hydrogel Solution Kit (C1310X, Logos Biosystems) for 3 days at 4 °C, allowing diffusion of the hydrogel solution into the tissue. Polymerization of solution was carried in a Logos Polymerization (C20001, Logos Biosystem) system at 37 °C for 3 h. (SDS-Clearing solution: For 2 L of 4% SDS solution used 24.73 g of boric acid (Sigma B7660 or Thermofisher B3750), 80 g of sodium dodecyl sulfate (Brunschwig 45900-0010, Acros 419530010 or Sigma L3771), in dH$_2$O, final solution pH 8.5).

After two washes of 30 min in PBS, samples were immersed in a SDS based clearing solution and left at 37 °C for 48 h. Once cleared, tissue was washed twice in PBS-TritonX 0.1% and then placed in a Histodenz© based-refractive index-matching solution (Histodenz Sigma D22158, PB + Tween + NaN$_3$ pH 7.5 solution, 0.1% Tween-20, 0.01% NaN$_3$, in 0.02 M phosphate buffer, final solution pH 7.5)

Imaging was performed with a home-built mesoscale single-plane illumination microscope; complete description of the mesoSPIM microscope is available here: (Voigt et al.[47]). Briefly, using one of the two excitation paths, the sample was

excited with 488 and 561 nm laser. The beam waist was scanned using electrically tunable lenses (ETL, Optotune EL-16-40-5D-TC-L) synchronized with the rolling shutter of the sCMOS camera. This produced a uniform axial resolution across the field-of-vie7w (FOV) of 5 μm. GFP autofluorescence signal was filtered with 530/43 nm, 593/40 nm bandpass filter (BrightLine HC, AHF). Z-stacks were acquired at 5 μm spacing with a zoom set at ×1.25 resulting in an in-plane pixel size of 5.26 μm.

Images were pre-processed to subtract the background and autofluorescence signal using the 561 nm excitation channel and subsequent normalization and filtering of the images were performed with the Amira 2019.4 software. 3D videos and images were captured using the Imaris 9.5 software.

*Tissue collection and cell preparation for FACS-sorting.* Forelimb and hindlimb buds from embryos with 40–45 somites (E12.5) were dissected in cold PBS solution. After PBS removal, a single cell suspension was achieved by incubating the limb buds in 400uL Trypsin-EDTA (Thermo Fischer Scientific, 25300062) for 12′ at 37 °C in a Thermomixer with a resuspension step at the 6′ mark. After blocking with one volume of 5% BSA (Sigma Aldrich, A7906-100G), cells were passed through a 40 μm cell strainer for further tissue disruption and another volume of 5% BSA was added to the cell strainer to pass leftover cells. Cells were then centrifuged at 400 × g for 5′ at 4 °C and, after discarding the supernatant, they were resuspended in 1% BSA for cell sorting. 5 mM of NaButyrate were added to the BSA when planning for subsequent fixation for H3K27Ac-ChIP.

*Proliferation and apoptosis analyses.* After tissue collection and cell dissociation, apoptotic cells were identified through Annexin V staining (Invitrogen, R37177). Following the manufacturer's instructions, two replicates of $2 \times 10^5$ cells were resuspended in the kit's binding buffer and 1 drop of Annexin V stain was added per $1 \times 10^5$ cells and left at room temperature incubation for 15′. Apoptotic phases were then determined by flow cytometry analysis. For cell cycle analysis we used two replicates of $2 \times 10^5$ cells, where DNA was stained with Hoechst-33342 dye (Abcam, ab228551) used to a final concentration of 5 μg/ml in a 1% BSA media. Cells were left for a 30′ incubation in a 37 °C water bath. Flow cytometry was then used to determine cell cycle stage. Both experiments were performed on a BDLSR Fortessa analyser and data was then processed using the FlowJo™ Software (10.6.1).

*Cell sorting.* Cell populations were isolated using fluorescent-activated cell sorting (FACS) using the Beckman Coulter MoFlo Astrios with GFP laser (excitation wavelength 488 nm). Initial FSC/SCC was set between 30/40 and 210/240 to exclude debris. After removal of dead cells with Draq7 dye and removal of doublets, following standard protocol, cells were gated for sorting as can be seen in Fig. S1A. As a control, a non-GFP expressing tissue (forelimbs isolated from the same E12.5 embryos) was used to determine the gating of the GFP− fraction of the samples to sort. When multiple cell sortings were needed, gating was done in accordance to previous samples to ensure non-variability in GFP intensity. Flow cytometry analysis to obtain GFP histograms was performed with the FlowJo™ Software (version 10.6.1).

*Cell processing for ChIP-seq and Capture-HiC.* After sorting, cells were centrifuged for 5′ at 400 × g at 4 °C and supernatant was discarded. Cells for ChIP-seq and Capture-HiC were resuspended in 10% FCS/PBS and fixed in 1% formaldehyde for ChIP and 2% for Capture-HiC at room temperature. The fixation was blocked by the addition of 1.25 M glycine, cells were isolated by centrifugation (1000 × g, at 4 °C for 8′), resuspended in cold lysis buffer (10 mM Tris, pH 7.5, 10 mM NaCl, 5 mM MgCl₂, 0.1 mM EGTA, Protease Inhibitor (Roche, 04693159001)) and incubated on ice for 10′ to isolate the cell nuclei. The nuclei were isolated by centrifugation (1000 × g, at 4 °C for 3′), washed in cold 1× PBS, centrifuged again (1000 × g, at 4 °C for 1′) and stored frozen at −80 °C after removal of the PBS supernatant.

*RNA-seq Cell processing and library preparation.* After sorting, cells were centrifuged for 5′ at 400 × g at 4 °C, supernatant was discarded and cells frozen at −80 °C. At least two biological replicates of $1.5 \times 10^5$ cells each were used to extract total RNA using the RNeasy Micro Kit (QIAGEN, ID:74004) following manufacturer's instructions and then stored frozen at −80 °C. Total RNA was quantified with a Qubit (fluorimeter from Life Technologies) and RNA integrity assessed with a Bioanalyzer (Agilent Technologies). The SMART-Seq v4 kit from Clontech was used for the reverse transcription and cDNA amplification according to the manufacturer's specifications, starting with 5 ng of total RNA as input. 200 pg of cDNA were used for library preparation using the Nextera XT kit from Illumina. Library molarity and quality was assessed with the Qubit and Tapestation using a DNA High sensitivity chip (Agilent Technologies). Libraries were pooled at 2 nM and loaded for clustering on a Single-read Illumina Flow cell for an average of 35 mio reads/library. Reads of 50 bases were generated using the TruSeq SBS chemistry on an Illumina HiSeq 4000 sequencer.

*ChIP-seq and library preparation.* $5 \times 10^5$ fixed nuclei were sonicated to a 200–500 bp length with the Bioruptor Pico sonicator (Diagenode). H3K27Ac ChIP (Diagenode C15410174) was performed as previously described[48,49], using 1/500

dilution of the antibody, with the addition of 5 mM of Na-Butyrate to all buffers. Libraries were then prepared following the Illumina ChIP TruSeq protocol and sequenced as 50 bp single-end reads on a illumina HiSeq 4000. Libraries were prepared starting with below <10 ng quantities of ChIP-enriched DNA as starting material and processed with the Illumina TruSeq ChIP kit according to manufacturer specifications. Libraries were validated on a Tapestation 2200 (Agilent) and a Qubit fluorimeter (Invitrogen – Thermofisher Scientific). Libraries were pooled at 2 nM and loaded for clustering on a Single-read Illumina Flow cell. Reads of 50 bases were generated using the TruSeq SBS chemistry on an Illumina HiSeq 4000 sequencer.

*Capture-HiC and library preparation.* 3C libraries were prepared as previously described[48]. Briefly, at least $1 \times 10^6$ fixed cells were digested using the DpnII restriction enzyme (NEB, R0543M). Chromatin was re-ligated with T4 ligase (Thermo Fisher Scientific), de-crosslinked and precipitated. To check the validity of the experiment, 500 ng of re-ligated DNA were loaded on a 1% gel along with undigested and digested controls. 3C libraries were sheared and adapters ligated to the libraries according to the manufacturer's instructions for Illumina sequencing (Agilent). Pre-amplified libraries were hybridized to the custom-designed SureSelect beads (chr13: 54,000,001–57,300,000)[11] and indexed for sequencing (50–100 bp paired-end) following the manufacturer's instructions (Agilent). Enriched libraries were pooled at 2 nM and loaded for clustering on a Paired-End Illumina Flow cell for an average of 215 mio reads/library. Reads of 100 bases were generated using the TruSeq SBS chemistry on an Illumina HiSeq 4000 sequencer.

### ChIP-seq, RNA-seq and Capture-HiC data analyses

*ChIP-seq.* Single-end reads were mapped to the reference genome NCBI37/mm9 using Bowtie2 version 2.3.4.2[50], filtered for mapping quality $q \geq 25$ and duplicates were removed with SAMtools 1.9. Reads were extended to 250 bp and scaled (1 million/total of unique reads) to produce coverage tracks using genomecov of BEDTools/2.28.0-fecbf4e3. BigWig files were produced using bedGraphToBigWig version 4 and visualized in the UCSC genome browser.

*RNA-seq.* Single-end reads were mapped to the mm9 reference genome using STAR mapper version 2.5.2a with default settings. Further processing was done according to ref. [48]. BigWig files were visualized in the UCSC genome browser. Counting was done using R version 3.6.2 and differential expression was analyzed through the "DEseq2" R package (version 3.14). The DEseq2 R package was also used to produce heatmaps by subtracting from each gene value per condition, given by vst, the mean value of all conditions. Genes were picked according to adjusted p-value, all being significantly differentially expressed between conditions. *Pitx1* fold enrichment between wildtype GFP+ and GFP− and between GFP−, GFP+− and GFP++ populations was calculated using Deseq2's normalization by size factor with the addition of a 0.5 pseudocount to aid data visualization. The Wald-test was used to examine the differential across samples. The p-values were adjusted for multiple testing with the FDR/Benjamin–Hochberg (BH) method and analysis was performed with at least two biological replicates. Expression heatmaps were generated for non-mesenchyme satellite and mesenchymal markers as defined in Supplementary Dataset S1. For visualization reasons, *Ccr5*, *Cldn5*, and *Col2a1* were added as sub-cluster markers (endothelium immune and condensation) and the forelimb-specific marker *Tbx5* was removed from the marker list. Moreover, genes with expression less or equal to 1 RPKM in all 8 samples (GFP+ wildtype: replicate 1 and 2; GFP− wildtype: replicate 1 and 2, GFP+ mutant: replicate 1 and 2; GFP− mutant: replicate 1 and 2) were removed from the analysis. For the GFP− specific heatmap, we additionally removed all genes with less or equal to 1 RPKM in all 4 GFP− samples. The color of the expression heatmap corresponds to the z-score transformed RPKM values, using the mean and standard deviation per gene based on all 8 samples. Log2FC was calculated by averaging replicates RPKM for each datasets and dividing *Pitx1*$^{GFP}$ and *Pitx1*$^{GFP;\Delta Pen}$ values.

*Capture-HiC and virtual 4C.* Paired-end reads from sequencing were mapped to the reference genome NCBI37/mm9 using with Bowtie2 version 2.3.4.2[50] and further filtered and deduplicated using HiCUP version 0.6.1. When replicates were available, these were pooled through catenation (-cat in Python 2.7.11) before HiCUP analysis. Valid and unique di-tags were filtered and further processed with Juicer tools version 1.9.9 to produce binned contact maps from valid read pairs with MAPQ ≥ 30 and maps were normalized using Knights and Ruiz matrix balancing, considering only the genomic region chr13: 54,000,001–57,300,000[51–53]. After KR normalization, maps were exported at 5 kb resolution. Subtraction maps were produced from the KR normalized maps and scaled together across their subdiagonals. C-HiC maps were visualized as heatmaps, where contacts above the 99thpercentile were truncated for visualization purposes. Further details about data processing can be accessed at ref. [11]. Virtual 4C profiles were generated from the filtered hicup.bam files used also for Capture-HiC analysis. The viewpoint for the *Pitx1* promoter was set at coordinates chr13:55,930,001–55,940,000 (10 kb bin) and contact analysis was performed over the entire genomic region considered for Capture-HiC (chr13: 54,000,001–57,300,000). A contact pair is considered when one interaction fragment is in the viewpoint and its pair mate is outside of it. The

interaction profile was smoothed by averaging over 5 kb intervals and was produced as a bedgraph file.

## Single-cell data analyses

*Processing of sequenced reads.* Demultiplexing, alignment, filtering barcode, and UMI counting was performed with 10× Genomics Cell Ranger software (version 3.0.2) following manufacture's recommendations, default settings and mm10 reference genome (version 3.0.0, provided by 10X Genomics, downloaded in 2019). Cell Ranger outputs files for each dataset were processed using the velocyto run10x shortcut from velocyto.py tool[24] (version 0.17.17) to generate a loom file for each sample, using as reference genome the one provided by 10× Genomics and the UCSC genome browser repeat masker.gtf file, to mask expressed repetitive elements. Each loom matrix, containing spliced/unspliced/ambiguous reads, was individually imported in R (version 3.6.2) with the Read Velocity function from the Seurat Wrappers package (version 0.2.0). In parallel, feature filtered output matrices obtained from Cell Ranger were individually loaded into R through the Read10X function of the Seurat package (version 3.2.0[54]). Then, we combined the spliced, unspliced, ambiguous, and RNA feature data in a single matrix for each dataset. Subsequently each matrix was transformed into a Seurat object using Seurat package. Therefore, for each sample we obtained for each sample a single Seurat object comprehend by four assays, three of them (spliced, unspliced and ambiguous) were used for downstream RNA velocities estimations and the RNA feature assay was used for downstream gene expression analysis between the samples, as described below.

*Quality control and filtering.* Quality control and pre-processing of each Seurat object of our eight samples was performed attending to the following criteria. Cells expressing less than 200 genes were excluded. Additionally, we calculated the reads that mapped to the mitochondrial genome and we filtered out the cells with a mitochondrial content higher than 15%, since high levels of mitochondrial mRNA has been associated to death cells. Also, we excluded cells with a mitochondrial content lower than 1%, since we observed that these belong, in our datasets, to blood cells probably coming from the dissection protocol.

*Individual dataset normalization, scaling, and dimensional reduction.* After filtering, one by one we normalized the eight datasets following the default Seurat parameters for the LogNormalize method and applying it only to the RNA features assay. We next scaled it by applying a linear transformation and we calculated the most variable features individually for downstream analysis, using standard Seurat parameters. Scaled data were then used for principal component analysis (PCA), we used the 50 PCs established by default, and non-linear dimensional reduction by Uniform Manifold Approximation Projection (UMAP[55]), we used 1:50 dims as input.

*Cell doublet identification.* Pre-process and normalized datasets were individually screened for detection of putative doublet cells. Doublets in each dataset were also excluded using DoubletFinder R package (version 2.0.2)[56] as described in https://github.com/chris-mcginnis-ucsf/DoubletFinder. The doublet rate (nExp parameter) used was estimated from the number of cells captured and it is as follows: $Pitx1^{+/+}$ Hindlimb replicate 1, nExp = 106; $Pitx1^{+/+}$ Hindlimb replicate 2, nExp = 123; $Pitx1^{+/+}$ Forelimb replicate 1, nExp = 97; $Pitx1^{+/+}$ Forelimb replicate 2, nExp = 116; $Pitx1^{-/-}$ Hindlimb replicate 1, nExp = 104; $Pitx1^{-/-}$ Hindlimb replicate 2, nExp = 122; $Pitx1^{Pen-/Pen-}$ Hindlimb replicate 1, nExp = 118; $Pitx1^{Pen-/Pen-}$ Hindlimb replicate 2, nExp = 116. The pK parameter was calculated following the strategy defined by ref. [56] and is as follow: $Pitx1^{+/+}$ Hindlimb replicate 1, pK = 0.12; $Pitx1^{+/+}$ Hindlimb replicate 2, pK = 0.005; $Pitx1^{+/+}$ Forelimb replicate 1, pK = 0.09; $Pitx1^{+/+}$ Forelimb replicate 2, pK = 0.04; $Pitx1^{-/-}$ Hindlimb replicate 1, pK = 0.04; $Pitx1^{-/-}$ Hindlimb replicate 2, pK = 0.01; $Pitx1^{Pen-/Pen-}$ Hindlimb replicate 1, pK = 0.005; $Pitx1^{Pen-/Pen-}$ Hindlimb replicate 2, pK = 0.005. After filtering, we kept for downstream analysis the following number of cells for each dataset: $Pitx1^{+/+}$ Hindlimb replicate 1, 4143 cells; $Pitx1^{+/+}$ Hindlimb replicate 2, 4816 cells; $Pitx1^{+/+}$ Forelimb replicate 1, 3802 cells; $Pitx1^{+/+}$ Forelimb replicate 2, 4521 cells; $Pitx1^{-/-}$ Hindlimb replicate 1, 4049 cells; $Pitx1^{-/-}$ Hindlimb replicate 2, 4745cells; $Pitx1^{Pen-/Pen-}$ Hindlimb replicate 1, 4600 cells; $Pitx1^{Pen-/Pen-}$ Hindlimb replicate 2, 4518 cells.

*Merge of all datasets and normalization.* Once each dataset was individually filtered and doublets were removed, all datasets were merged in a unique Seurat object without performing integration to execute an ensemble downstream analysis of the eight datasets. No batch effect was observed later on in this merged dataset. A new column to the Seurat object metadata was added to label replicates of the same tissue and animal model with the same name for downstream analysis. Therefore the cells of $Pitx1^{+/+}$ Hindlimb, replicate 1, and replicate 2 were labeled as $Pitx1^{+/+}$ Hindlimb, the same logic was applied to the rest of the samples. Subsequently, we normalized our new and unique Seurat object applying the SCTransform normalization protocol[57], with default parameters, over the spliced assay.

*Cell-cycle scoring and regression.* Since from the individual analysis of our dataset we observed a part of the variance was explained by cell-cycle genes, we examine

cell-cycle variation in the merged dataset. To do so we assigned to each cell a score based on its expression of a pre-determined list of cell cycle gene markers, following the strategy defined by[58] and by applying CellCycleScoring function implemented in Seurat. Subsequently, the evaluation of this results, we decided to regress out the cell-cycle heterogeneity. Therefore, we applied to our merged object the SCTransform normalization method, using the spliced assay as source, and adding to the default settings the cell-cycle calculated scores (S.Score and G2M.Scores) as variables to regressed. Cell Cycle classification was later on used to estimate cell cycle proportions on each cluster.

*Clustering.* After cell-cycle regression, cells were clustered using standard steps of the SCTransform Seurat workflow. Briefly, PCA (npcs = 50), UMAP (dims = 1:50) and nearest neighbors of each cell were calculated. Clusters were determined using Seurat FindClusters function with default parameters and a resolution of 0.2, in that way 10 clusters were defined. Identification of clusters identity was done by calculating the expression difference of each gene between each cluster and the rest of the clusters using the FindConservedMarkers function. We applied this function to each cluster (ident.1) using default parameters, only.pos = TRUE and setting as grouping variable the limb identity of the datasets, in that way we obtained a list of markers for each cluster independent of the limb sample. Clusters with similar marker were combined, therefore we finally worked with 5 clusters (Fig. 1B): the mesenchyme (that contains 5 out of the 10 clusters), the epithelium (formed by 2 out of 10), the immune cell cluster, the muscle and the endothelium clusters (composed by only 1 cluster each). We confirmed the expected identity markers were present in the new clustering by running the FindMarkers function with the following parameters logfc.threshold = 0.7; pseudocount.use = 0; only.pos = TRUE; min.diff.pct = 0.15 and all other default parameters (Supplementary Dataset S1).

*Subsetting and re-clustering.* Since the interest of this work was focus on the populations that in a wildtype hindlimb express $Pitx1$ (Fig. 1C), we subsetted the mesenchyme cluster. To have a better insight on the different cell-types that integrate it, we re-cluster the mesenchyme cluster. To do so, UMAP embedding was calculated with the following parameters: dims = c(1:10), n.neighbors = 15L, min.dist = 0.01, metric = "cosine", spread = 0.5, all other parameters were default. Cluster resolution after finding neighbors was established at 0.4 to reveal sub-populations. We observed 9 mesenchyme subpopulations (Fig. 3A) that we named according to their identity genes. Identity markers were found using FindMarkers on the RNA assay, setting logfc.threshold = 0.3, pseudocount = 0, min.diff.pct = 0.1, only.pos = TRUE and all other parameters as default (Supplementary Dataset S1).

*Differential expression analysis.* To perform $Pitx1^{+/+}$ Hindlimb vs $Pitx1^{Pen-/Pen-}$ Hindlimb differential expression analysis in the mesenchyme cluster and in each one of the nine mesenchyme clusters we used the FindMarkers function on the RNA assay. For whole mesenchyme analysis $Pitx1^{Pen-/Pen-}$ Hindlimb was set up as ident.1 and $Pitx1^{+/+}$ Hindlimb as ident.2. To determine the differentially expressed genes between each dataset in each mesenchyme cluster, we created a new column in the metadata slot that contains both cluster and dataset information. Then, this column was set as new identity and differential expression analysis was run using as ident.1: ICT_Pitx1$^{Pen-/Pen-}$, Ms_Pitx1$^{Pen-/Pen-}$, TP_Pitx1$^{Pen-/Pen-}$, LDC_Pitx1$^{Pen-/Pen-}$, DPP_Pitx1$^{Pen-/Pen-}$, DP_Pitx1$^{Pen-/Pen-}$, PC_Pitx1$^{Pen-/Pen-}$, EDC_Pitx1$^{Pen-/Pen-}$, or PPP_Pitx1$^{Pen-/Pen-}$ and as ident.2: ICT_HL_Pitx1$^{+/+}$, Ms_HL_Pitx1$^{+/+}$, TP_HL_Pitx1$^{+/+}$, LDC_HL_Pitx1$^{+/+}$, DPP_HL_Pitx1$^{+/+}$, DP_HL_Pitx1$^{+/+}$, PC_HL_Pitx1$^{+/+}$, EDC_HL_Pitx1$^{+/+}$ or PPP_HL_Pitx1$^{+/+}$. All the other parameters as default except setting logfc.threshold = 0.2, pseudocount.use = 0 (Supplementary Dataset S5).

*RNA-velocity analysis.* As input data for the RNA-velocity analysis, we used the unspliced (pre-mature) and spliced (mature) abundances calculated for each replicate of our datasets as explained above (see in Methods, Processing of sequencing reads). To perform the RNA velocity analysis on the mesenchyme clusters of each dataset we subset the cells belonging to the 2 replicates. Therefore, we subset $Pitx1^{+/+}$ Hindlimb, $Pitx1^{+/+}$ Forelimb, $Pitx1^{Pen-/Pen-}$ Hindlimb, and $Pitx1^{-/-}$ Hindlimb individually. We also performed RNA-velocity analysis of all combined datasets. To perform proximal clusters and distal clusters analysis, we subset them separately following the criteria for proximal and distal cluster classification that is explained below in the Methods. Seurat objects from which we performed RNA-velocity analysis were saved as h5Seurat file using SeuratDisk package (version 0.0.0.9013) and exported to be used as input of Scvelo (version 0.2.2)[59] in Python (version 3.7.3). Then the standard protocol described in scVelo was followed. Standard parameters were used except npcs = 10 and n.neighbors = 15, to be the same that we used for the UMAP embedding in Seurat.

*Differential proportion analysis.* Statistical differential proportion analysis, to study the differences in clusters cell proportions between the different limb-type conditions, was performed in R using the source code published by[60] after generating the proportion tables in R. Null distribution was calculated using n = 100,000 and p = 0.1 as in the original reference. Pairwise comparisons were performed between the different condition tested.

*Proximal and distal cell classification*. Proximal, distal or NR attribute was given to each cluster based on its Shox2 and Hoxd13 expression. Therefore, ICT, TP, PPP, and PC clusters were classified as proximal clusters, DP, DPP, EDC, and LDC as distal ones. Meanwhile Ms cluster that express both markers were not classify to any of them. This classification was added to the Seurat object metadata and used in downstream analysis.

Pitx1 *density plot and cell classification by* Pitx1 *expression*. Pitx1 normalized expression values (using Seurat default LogNormalize method, using log1p), from the RNA assay of the all dataset merged Seurat Object, were extracted in a data frame. This data frame was used to create a density plot using ggplot2 package (version 3.3.2). From the overlay of Pitx1 density distributions in the $Pitx1^{+/+}$ Hindlimb and the $Pitx1^{Pen-/Pen-}$ Hindlimb samples we define the intersection point of 0.3 to classify cells in non/low-expressing and expressing cells. The second intersection point of 1.45 that subclassify these expressing cells in intermediate- and high- expressing cells was established based on the intersection of the $Pitx1^{+/+}$ Hindlimb proximal and distal cells (Fig. 2F). Therefore, we classified as non/low-expressing cells those with Pitx1 expression values <0.3, as intermediate-expressing those with Pitx1 expressing values between >0.3, <1.45 and as high-expressing cells those >1.45. This classification and Pitx1 expression values were added as new columns to the Seurat object metadata and used in downstream analysis.

**Reporting summary**. Further information on research design is available in the Nature Research Reporting Summary linked to this article.

## Data availability

Sequencing data are available at the GEO repository under the accession number "GSE168633". All other relevant data supporting the key findings of this study are available within the article and its Supplementary Information files or from the corresponding author upon reasonable request. Source data are provided with this paper.

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

## Acknowledgements

We thank Mylène Docquier, Brice Petit, and Christelle Barraclough from the iGE3 sequencing facility. We thank Jean-Pierre Aubry, Grégory Schneiter and Cécile Gameiro from the Flow Cytometry facility. We thank Nicolas Liaudet from the imaging facility and Stéphane Pàges and Laura Batti from Advanced Light Sheet Imaging Center (ALICe) at the Wyss Center for Bio and Neuroengineering, Geneva. We thank Leon Van Gurp, Sigmar Stricker, Pierre Fabre, Quentin Lo Giudice, Lucille Delisle, and Denis Duboule for discussions. We thank Michael Robson, Anna Ramisch, Simon Braun, and Christina Paliou for critical reading of the manuscript. We thank all lab members for discussions and critical reading of the manuscript. This study was supported by grants from the Swiss National Science Foundation (PP00P3_176802) and from the Boninchi Foundation to G.A.

## Author contributions

G.A., O.B., and R.R. conceived the project. R.R., O.B., R.P., and A.R. performed the ESC targeting and prepared the cells for aggregation. O.B. performed the Capture-HiC, ChIP-eq, and RNA-seq preparations and analyses. R.R. performed the scRNA-seq preparation and analyses. O.F. and F.T. established and performed tetraploid aggregations. G.A., R.R., and O.B. wrote the manuscript with input from the remaining authors.

## Competing interests

The authors declare no competing interests.

## Additional information

**Peer review information** *Nature Communications* thanks Chrissa Kioussi, Stephanie N. Oprescu, Vittorio Sartorelli and the other anonymous reviewer(s) for their contribution to the peer review this work. Peer reviewer reports are available.

