## [Peer Review File · Nature Communications]

Reviewers' Comments:

Reviewer #1:

Remarks to the Author:

The study by Rouco et al. addresses a very important issue concerning cell-type specific transcriptional enhancer activities and their interactions during embryonic tissue morphogenesis. They study this by analyzing the enhancers regulating Pitx1 expression in hindlimb buds. In particular, the effects of germline deletion of one of several enhancers that regulate Pitx1 expression on transcript levels, enhancer activities and chromatin structure in hindlimb bud mesenchymal cell populations are analyzed. Single cell analysis is combined with the analysis of FACS sorted cell populations to determine Pitx1 expression levels, enhancer activities and chromatin structure in mesenchymal cell populations. This establishes differences in transcript levels, chromatin activity and structure between Pitx1 cell populations that express high, mild and low transcript levels. Genetic inactivation of the Pen enhancer results in a decrease in high and concurrent increase in low expressing cells; the later preferentially the proximal hindlimb bud mesenchyme. In addition, the analysis shows that the Pen deletion is paralleled by changes in chromatin interactions and activities of the other Pitx1 enhancers (H3K23ac modifications). They propose that these changes underlie the resulting clubfoot phenotype in Pen-deficient mice. The results are overall convincing and use state-of-the art technology such that a significantly revised version should be suitable for the broad readership of Nature Communication.

Issues to be addressed for revision:

1. The authors use scRNA-seq analysis in combination with marker genes to identify both proximal and distal mesenchymal cells clusters. While the data analysis shown is state-of-the art, the correspondence of a significant number of the mesenchymal populations with previously identified key limb bud cell populations such as early and late progenitors, osteochondrogenic progenitors (OCPs) and chondrocytes is not clear (in contrast to e.g. tendons). Furthermore, there is no experimental evidence the proximal and distal progenitor populations identified are really proliferating cells. It would be important to establish if they express known cell-cycle markers at higher levels. What are the cells in the early and late digit condensation populations - are these OCPs and/or chondrocytes at different stages?
2. Linked to the mesenchymal cell population analysis: are there any changes in apoptosis (or proliferation) when comparing wt and Pen-mutant GFP+ and GFP- mesenchymal cell populations. This would be important to know as it might contribute to the observed shift from high to low expressing cells. Such analysis is straightforward as FACS can be combined with markers for cell death and cell cycle.
3. Even after consulting the methods section it was not clear to this reviewer how the velocity analysis (Fig. 2E) allows the authors to arrive at conclusions concerning cell-lineage. This velocity analysis needs to be described in more detail.
4. This reviewer does not understand what the authors mean with the term "mild"-expressing cells as it is not a common term to describe transcript levels. Does it refer to cells expressing intermediate levels of Pitx1? In addition, it should be stated what is the range of transcript levels for high, intermediate (mild), low and no expression. This is important as global Pitx1 transcript levels reduced by ~30% in Pen-deficient hindlimb buds. Does the intermediate (mild) correspond to a ~30% (= average) reduction in the specific cell populations?
5. The description of the results (page 10-12) shown in Fig. 4, Fig. S5 and S6 must be extended and significantly more detail given. Its current state is not well structured and not easy to understand for non-specialists. In particular, the data shown in Fig. S5 are very important to the general understanding of the findings and should be incorporated into a main figure. The flow and description of this section must be improved as it describes one of the key findings of this study.
6. Last paragraph on page 12: the changes in Tbx4 expression are difficult to link cell proliferation without analyzing the proliferation kinetics of these cell populations (see point 2 above).
7. This reviewer is also confused by the term "escaping" high expressing cells. As the authors show these Pen-deficient high expressing cells retain Pitx1 expression at levels similar to their wild-type counterparts. This points to cis-regulatory redundancy, which is substantiated by the data shown in Fig. 5.
8. Correlation of the observed population-specific alterations in Pitx1 expression with the club foot phenotype in Pen-deficient mice. The authors only discuss the alterations affecting the ICT

population, while is equally likely that the heterochrony observed in several of the mesenchymal cell populations could collectively cause the phenotype as coordinated expansion and interactions among different mesenchymal cell populations is required for normal limb morphogenesis.
9. There is misspellings in the text that must be corrected.

Reviewer #2:

Remarks to the Author:

In this report, Rouco and colleagues defined the contribution of Pen1, one of the four known Pitx1 enhancers, in establishing the epigenetically and structurally active Pitx1 regulatory landscape. More specifically, the authors characterized how Pen1 contributes to Pitx1 activity in specific cell populations. This is very well designed and executed research, and very much needed to understand the gene regulation of hindlimb formation. Several points need further attention and clarification.

Introduction

1. The publications that describe the Pitx2 KO hindlimb phenotype are incomplete.
2. A short summary of the function of all enhancers (Pen, Pit, RA4, PeIB) and the rationale for focusing on Pen1 should be considered, including their transcriptional active state during development.

Results

1. The authors appear to use mice in Fig 1A only for data in Fig 1B and 1C. The fact that Pitx1 is in mesenchyme is of interest but it's unclear what the purpose of these genotypes in A is. A scRNA analysis of a wt animal would have sufficed to set data shown in Fig 1B and 1C. There appears to be no other mention of the use of these three mice in the manuscript. It would be of interest to know how populations shift or change with the wt vs Pitx1^{-/-} or wt vs Pitx1D^{Pen} comparisons.
2. The hindlimb consists of 90% Prrx1⁺ mesenchymal cells (line 95). The 8% of Pitx1 active cells include mesenchymal, immune, endothelium, epithelium, and muscle origin (line 134). The latter information is not well supported by the data provided in Figure 1F and seems like that the numbers do not match in the two areas, please correct.
3. The authors did not explain the presence of the high H3K27ac activity upstream of Pen, close to the H2afy locus (Fig 3D).
4. More outcomes after comparing Pitx1^{+/+} vs Pitx1GFP and similarly Pitx1Pen^{-/}Pen⁻ vs Pitx1GFP:ΔPen transcriptomes should be included.
5. Figure 4 is difficult to follow since analysis has been done in different sets of biopsies, whole hindlimb, and flow-sorted Pitx1⁺ hindlimb cells. In addition, the authors should explain the baseline on the expression levels in Fig4D.
6. Minguillon et al 2004 propose that Pitx1 and a combinatorial code of "rostral" Hox genes trigger the Tbx4 expression. The authors should determine the Hox cluster alterations in the absence of Pen1.

Discussion

1. The potential function of the other enhancers should be included.
2. The developmental/evolutionary conservation of the Pen1 enhancer should be included.

Minor points

A brief description of the supplementary tables would facilitate the data interpretation.

Reviewer #3:

Remarks to the Author:

(I) Summary/Findings

The authors combine both genetic mouse models and high throughput sequencing techniques to outline the role of the Pitx1 enhancer and the local 3D chromatin architecture in hindlimb development. The authors provide evidence to show that indeed the Pitx1 regulatory locus functions as an entire unit and is not uniquely and fully dependent on the Pen enhancer. This data is supported by their H3K27ac and 3D chromatin conformation assays.

These results nicely outline the effects of loss of a known Pitx1 enhancer. The authors find that loss of Pen leads to an increase in the number progenitor cells as evidenced by their scRNA-seq data at the expense of proximal, Pitx1 high expressing cells. Using their newly generated mouse model that tracks the expression of Pitx1 via GFP, the authors are able to confirm that indeed loss of Pen leads to reduced Pitx1 expression (as tracked by GFP), leading to an increase in the proportion of GFP-lower expressing cells that were shown to exhibit H3K27ac at known distal genes. Interestingly, loss of Pitx1 leads to an increase in progenitor cells (so fewer proximal and distal differentiated cells), while loss of Pen leads to a similar phenotype but only in the proximal cell clusters, highlighting the role of the Pen enhancer in proximal cells. The increase in progenitor cells observed in the Pitx1^{-/-} and Pen^{-/-} mice may thus contribute to the clubfoot phenotype, which may be the result of impaired differentiation of cells that contribute to hindlimb structures.

Overall, the authors characterize the role of the Pen enhancer during limb development. Indeed, this is important to further our understanding of enhancer functions, specifically when establishing both gene expression and chromatin landscapes during development and thus would provide significant knowledge to the field.

(II) Major comments

1. The authors suggest that the differences in Pitx1 expression is associated with a progressive increase in its cis-regulatory landscape rather than the activity of individual enhancers. What might modulate the expression of Pitx1 expression? Could there be other chromatin conformations that are not at the analyzed locus that could modulate the expression of Pitx1?
2. Besides Tbx4, are there other genes that Pitx1 regulates whose expression is altered that might lead to the increased numbers of distal cells? Can the authors look into the expression of Pitx1-regulated genes to try and link the observed phenotype with the bimodal levels of Pitx1?
3. Are there any other loci involved in proximal limb development that show altered H3K27ac in the GFP⁺ vs GFP⁺⁺ (or in the Pen^{-/-} mice)? As the authors remark, a failure to lead to some threshold of gene expression may lead to the observed clubfoot phenotype, but more analysis should be added to support this claim.
4. Authors suggest that the clubfoot phenotype may be the result of increased progenitor cells, however this claim is not clearly supported. Can the authors identify more targets (such as Tbx4 or others) and use an in vitro model to evaluate whether restoration of these targets is able to support differentiation of a proximal cell phenotype. Or are the progenitor cells in the Pen^{-/-} mice more poorly differentiated at later stages of development? Perhaps a time-course experiment (ideally with scRNA-seq data if feasible) would help to better link these data with the clubfoot phenotype.
5. The authors highlight reduced number of Tbx4 expressing cells in Pen^{-/-} mice being similar to loss of Pitx1^{-/-} and suggest that loss of Pitx1 expression by deletion of the enhancer might be mediated by Tbx4. Are the Tbx4-negative cells that result as a loss of Pen or Pitx1 transcriptionally similar? Can the authors show a UMAP plot of the scRNA-seq data to highlight the changes in the proportions of cells (to better compare and visualize the difference or similarities between the two genotypes)?
6. Did the authors detect any interchromosomal contacts or other long-range contacts that might be altered in the Pitx1^{-/-} and Pen^{-/-} models?
7. The cells that are able to maintain the GFP⁺ expression under loss of Pen maintain similar RNA profiles; which suggests there may also be other factors that contribute to the observed local chromatin conformation that supports active Pitx1 transcription. Can the authors speculate on what some of these factors might be?

(III) Minor comments

1. Figure 1B: Should include UMAP colored by sample/genotype.
2. Authors state that in GFP- cells deleted for Pen, there was ectopic acetylation of Pitx1; can the authors look at the expression of GFP as well to see if its significantly enriched in this cluster to further support their claim that this is due to the relocation of GFP- cells in the gating strategy.
3. Generally, more UMAP plots need to be added to show the differences in the cell type distribution across the various genotypes.

Reviewer #4:

Remarks to the Author:

This manuscript reports the characterization of Pitx1 regulatory regions in sorted cells derived from the mouse limb. Deletion of the Pen enhancer (Pitx1 Pen-/Pen-) resulted in decreased number of cells expressing high Pitx1 levels and increased cells expressing low Pitx1 levels. scRNA-seq allowed the identification of several cell clusters, expressing different Pitx1 levels. In cells not expressing Pitx1, the Pen enhancer contacted the repressed Neurog1 gene. Pitx1 Pen-/Pen- cells also displayed reduced Tbx4 expression. Some Pitx1 Pen-/Pen- cells (high-expressing escaping cells) expressed high Pitx1 level. H3K27 acetylation landscape was indistinguishable in WT GFP+ (high expressing cells) and Pitx1 Pen-/Pen- (high-expressing escaping cells). 3D conformation of Pitx1 Pen-/Pen- (high-expressing escaping cells) was different from that adopted by WT cells, indicating that escaping cells do not require Pen to establish an active 3D conformation. Slight reduction of contact between the Pitx1 promoter and another enhancer (Pit/RA4) was noted.

The experiments are well conducted and controlled and the findings of interest. There are few issues that requiring clarification.

1. Figure 1G. A higher-definition map would allow better visualization of the contacts established by Pitx1 and its regulatory elements with Neurog1.
2. Figure S7. How does the Pen deletion affect Tbx4? Is this a direct or indirect effect?
3. While Pen enhancer is clearly important to regulate Pitx1 expression, the lack of 3D active contacts and similar H3K27ac landscape in Pitx1 Pen-/Pen- high-expressing escaping cells suggest its dispensability. Perhaps subpopulation(s) of mesenchymal cells, not identified in this study, employ a distinct regulatory network to regulate Pitx1 expression. The authors may want to speculate on this point.

Reviewer #1 (Remarks to the Author):

The study by Rouco et al. addresses a very important issue concerning cell-type specific transcriptional enhancer activities and their interactions during embryonic tissue morphogenesis. They study this by analyzing the enhancers regulating Pitx1 expression in hindlimb buds. In particular, the effects of germline deletion of one of several enhancers that regulate Pitx1 expression on transcript levels, enhancer activities and chromatin structure in hindlimb bud mesenchymal cell populations are analyzed. Single cell analysis is combined with the analysis of FACS sorted cell populations to determine Pitx1 expression levels, enhancer activities and chromatin structure in mesenchymal cell populations. This establishes differences in transcript levels, chromatin activity and structure between Pitx1 cell populations that express high, mild and low transcript levels. Genetic inactivation of the Pen enhancer results in a decrease in high and concurrent increase in low expressing cells; the later preferentially the proximal hindlimb bud mesenchyme. In addition, the analysis shows that the Pen deletion is paralleled by changes in chromatin interactions and activities of the other Pitx1 enhancers (H3K23ac modifications). They propose that these changes underlie the resulting clubfoot phenotype in Pen-deficient mice. The results are overall convincing and use state-of-the art technology such that a significantly revised version should be suitable for the broad readership of Nature Communication.

We would like to thank this reviewer for their appreciation of our work and for all the constructive comments that we have tried to answer below.

Issues to be addressed for revision:

1. The authors use scRNA-seq analysis in combination with marker genes to identify both proximal and distal mesenchymal cells clusters. While the data analysis shown is state-of-the art, the correspondence of a significant number of the mesenchymal populations with previously identified key limb bud cell populations such as early and late progenitors, osteochondrogenic progenitors (OCPs) and chondrocytes is not clear (in contrast to e.g. tendons). Furthermore, there is no experimental evidence the proximal and distal progenitor populations identified are really proliferating cells. It would be important to establish if they express known cell-cycle markers at higher levels. What are the cells in the early and late digit condensation populations - are these OCPs and/or chondrocytes at different stages?

We thank the reviewer for this suggestion. Indeed, a detailed comparison with previously identified limb cell population will improve the coherence of the manuscript. We have therefore adapted the description of the clusters from line 161 to 184, we have focused our cluster definitions according to the following three points and added new analyses in a new figure S6 (see below):

- *First, with regard to the progenitor cells: all our progenitors are expressing Tbx2 and Tbx3 but not Sox9 suggesting that they are a late form of Limb Mesenchymal Progenitors (LMPs, Reinhardt et al., Development, 2019) (See Fig. S6B below). Distal Progenitors (DP) and Distal Proliferative Progenitors (DPP) express low level of Jag1, but high level Msx1 (See Fig. 2) further confirming them being late LMPs. Moreover, Proximal Proliferative Progenitors (PPP) don't express Jag1 but Alx4 (See Fig. S6B) and Irx5 (See Fig. 2) which are also markers of LMPs (Reinhardt et al., Development, 2019).*
- *Second, we show now in figure S6A (see below) that distal and proximal progenitors are mostly constituted of G2 and S-phase cells confirming their highly proliferative status. This is now referenced line 163 and 176.*
- *Finally, we have defined that Proximal Condensations (PC) and Late Digit Condensation (LDC) correspond to chondrocytes as they both express collagen genes Col2a1 and Col9a3 on top of Sox9 (See Fig. S6B). In contrast, Early Digit Condensation (EDC) corresponds to distal OCPs as they expressed Sox9 but not Col2a1 and Col9a3 (See Fig. S6B).*

2. Linked to the mesenchymal cell population analysis: are there any changes in apoptosis (or proliferation) when comparing wt and Pen-mutant GFP+ and GFP- mesenchymal cell populations. This would be important to know as it might contribute to the observed shift from high to low expressing cells. Such analysis is straightforward as FACS can be combined with markers for cell death and cell cycle.

As suggested by the reviewer, we have added a FACS analysis of cell proliferation and apoptosis.

- *First, we performed FACS analysis of wildtype GFP+ and GFP- hindlimb cells using Hoechst and Annexin V staining to measure proliferation and apoptosis, respectively. We display this in a new S4 figure (see below) and comment it in the result section at line 134: "Finally, as Pitx1 was previously associated with tissue outgrowth, we also assayed proliferative and apoptotic behaviors of GFP+ and GFP- cells (Duboc and Logan 2011). As suspected, we found that GFP+/Pitx1+ cells are slightly more proliferative and contain less apoptotic cells (Fig. S4A-B)."*

- *Second, we used the same approach on co-aggregated (same litter embryos) wildtype and Pen-deleted hindlimbs. We find that neither the overall proliferation nor the proportion of apoptotic is affected by the loss of the Pen enhancer. We believe that this data further reinforces the idea that a differentiation rather than a proliferation alteration underlies the changes in cell type composition of the limb. We have added this data in a new supplementary figure S13 (see below) and write in the manuscript line 310: "Finally, we measured if the Tbx4 expression loss was sufficient to alter cell proliferation and apoptosis, therefore changing the hindlimb cell type composition (Duboc and Logan 2011). Overall, we did not observe changes in either, suggesting that the observed loss of Tbx4 is not sufficient to alter cell proliferation and apoptosis, and that the induced phenotype takes origin from an independent mechanism (Fig. S13A-B)."*

3. Even after consulting the methods section it was not clear to this reviewer how the velocity analysis (Fig. 2E) allows the authors to arrive at conclusions concerning cell-lineage. This velocity analysis needs to be described in more detail.

We agree that a more extensive description is lacking from the current version of the manuscript. In a nutshell velocity uses the ratio between mature spliced transcript and introns-containing transcripts to link progenitors (more immature transcript of a given gene) and differentiated cells (more mature transcript of the same gene). Overall, it appears that proximal and distal clusters are not linked by velocity although they may both contribute to the mesopodium cluster. We have now increased the description in the main text and material and methods as well as added new velocity analysis of the mesenchyme for each sample/genotype to better highlight predicted differentiation directions. We write now line 185: "To better understand the links between the different clusters, we ran an RNA velocity analysis in the hindlimb dataset, which predicts cell lineage differentiation based on the dynamics of spliced (mature) versus unspliced (immature) mRNAs (Fig. 2E)(La Manno et al. 2018; Bergen et al. 2020)."

We also have made a separate analysis for proximal and distal mesenchyme clusters (Fig. 5C and 5D, see below), to study the delay in differentiation in the Pitx1^{Pen-/Pen-} and Pitx1^{-/-} compared to the wildtype limbs as we explained from line 286 on: "To test if these effects could be explained by a delayed differentiation of progenitor cells, we performed a velocity analysis on Pitx1^{-/-}, Pitx1^{Pen-/Pen-} and Pitx1^{+/+} limbs in proximal and distal cell clusters separately. In the proximal part of the hindlimb, we found in both mutants a predicted connection from PPP to ICT cells, suggesting an ongoing differentiation process (Fig. 5C). This connection was not present in Pitx1^{+/+} fore- and

hindlimbs, suggesting that the differentiation process was completed in these tissues. These findings are further supported by an increase of ICT marker genes (*Lum*, *Dcn* and *Kera*) in PPP cells of both mutants, suggesting that those cells have only partially adopted an ICT identity but still did not fully differentiate (Fig. S11). In contrast, the velocity analysis of distal clusters did not show any changes in *Pitx1^{Pen-/Pen-}* hindlimbs, in agreement with the unaltered proportion of distal cell clusters shown above (Fig. 5B, 5D). Finally, we observed in *Pitx1^{-/-}* hindlimbs an accumulation of distal progenitor cells and a loss of differentiated LDC cells suggesting a slower distal differentiation process in *Pitx1^{-/-}* hindlimbs.”

4. This reviewer does not understand what the authors mean with the term “mild”-expressing cells as it is not a common term to describe transcript levels. Does it refer to cells expressing intermediate levels of *Pitx1*? In addition, it should be stated what is the range of transcript levels for high, intermediate (mild), low and no expression. This is important as global *Pitx1* transcript levels reduced by ~30% in *Pen*-deficient hindlimb buds. Does the intermediate (mild) correspond to a ~30% (= average) reduction in the specific cell populations?

Accordingly, we have now changed “mild”-expressing for “intermediate”-expressing in the manuscript. Concerning the transcript range, we have now written line 195: “We then classified mesenchymal *Pitx1* expressing cells in three categories: non/low-expressing (transcription values ≤ 0.3 , 21 % of the hindlimb wildtype cells), intermediate-expressing (transcription >0.3 ; ≤ 1.45 , 40 % of wildtype cells), and high-expressing (transcription >1.45 , 39 % of wildtype cells) (Fig. 2F-G).”

Finally, to answer the last part of this comment, the intermediate expression level corresponds to a 4-fold reduced average expression from the high-expressing cells. Indeed, the x-axis displayed in these analyses is a natural logarithm of (*Pitx1* expression + 1).

5. The description of the results (page 10-12) shown in Fig. 4, Fig. S5 and S6 must be extended and significantly more detail given. Its current state is not well structured and not easy to understand for non-specialists. In particular, the data shown in Fig. S5 are very important to the general understanding of the findings and should be incorporated into a main figure. The flow and description of this section must be improved as it describes one of the key findings of this study.

We appreciate this comment and agree that this part of the manuscript appears to be an overload of information. To improve its readability, we have now split figure 4 and the associated text section into 2 new figures and 2 new text sections, respectively. The new figure 4 is now dedicated to *Pitx1* expression alteration in *Pen*-deleted hindlimbs. The new figure 5 is now dedicated to cellular alterations following the enhancer deletion. Moreover, the previous figure S5 has now been shifted into the new figure 4E, 4F. Finally, we believe to have improved the flow and description of the entire section especially since several new datasets arising from this and other reviewer comments have now complemented the new figure 5.

6. Last paragraph on page 12: the changes in Tbx4 expression are difficult to link cell proliferation without analyzing the proliferation kinetics of these cell populations (see point 2 above).

In response, after re-analyzing our single cell experiment and adding proliferation data (see point 2), we can now conclude that the loss of Tbx4 seen in the Pen-deleted animals is not sufficient to induce an effect on proliferation, at least at the stage we looked at. We now write in the manuscript line 310: “Finally, we measured if the Tbx4 expression loss was sufficient to alter cell proliferation and apoptosis, therefore changing the hindlimb cell type composition (Duboc and Logan 2011). Overall, we did not observe changes in either, suggesting that the observed loss of Tbx4 is not sufficient to alter cell proliferation and apoptosis, and that the induced phenotype takes origin from an independent mechanism (Fig. S13A-B).

7. This reviewer is also confused by the term “escaping” high expressing cells. As the authors show these Pen-deficient high expressing cells retain Pitx1 expression at levels similar to their wild-type counterparts. This points to cis-regulatory redundancy, which is substantiated by the data shown in Fig. 5.

We have now clarified what we mean by escaping cells. Indeed, we believe that the cis-regulatory redundancy is exactly the mechanism that enables these cells to “escape” a loss of expression in comparison to other cells of the limb and thus maintain high Pitx1 expression level. We now write line 334: “This suggests that these high expressing cells, that escape a loss-of-expression following the deletion of Pen, must display an adaptive mechanism to accommodate the Pen enhancer loss.”

8. Correlation of the observed population-specific alterations in Pitx1 expression with the club foot phenotype in Pen-deficient mice. The authors only discuss the alterations affecting the ICT population, while is equally likely that the heterochrony observed in several of the mesenchymal cell populations could collectively cause the phenotype as coordinated expansion and interactions among different mesenchymal cell populations is required for normal limb morphogenesis.

This is an important insight that we have now integrated in the discussion line 427: “An alternative hypothesis could be that the observed heterochrony in several of the mesenchymal cell populations could collectively cause the clubfoot since coordinated expansion and interactions among different mesenchymal cell populations are required for normal limb morphogenesis.” Nevertheless, we think that the hypothesis of muscle patterning defects cause by ICT loss is highly likely as supported by multiple evidence (Chevallier and Kieny 1982; Kardon et al. 2002; Kardon et al. 2003; Swinehart et al. 2013; Colasanto et al. 2016; Vallecillo-Garcia et al. 2017; Besse et al. 2020). For the reviewer’s discretion, we have produced preliminary data in the lab that show hindlimb defects in muscle patterning when the Pen enhancer is deleted (see the Myog In Situ Hybridisation below, altered muscle bundles are shown by a white arrows). Yet, as we believe a full characterization of this phenotype would require more evidence and would also be outside the scope of this manuscript, we prefer not to include this piece of data in the manuscript.

9. There is misspellings in the text that must be corrected.

We have now improved misspellings and grammar in the new version of the manuscript.

Reviewer #2 (Remarks to the Author):

In this report, Rouco and colleagues defined the contribution of Pen1, one of the four known Pitx1 enhancers, in establishing the epigenetically and structurally active Pitx1 regulatory landscape. More specifically, the authors characterized how Pen1 contributes to Pitx1 activity in specific cell populations. This is very well designed and executed research, and very much needed to understand the gene regulation of hindlimb formation. Several points need further attention and clarification.

We thank the reviewer for their appreciation of our work as well as for all the constructive suggestions made.

Introduction

1. The publications that describe the Pitx2 KO hindlimb phenotype are incomplete.

Accordingly, we have now updated the Pitx1 KO citations line 58 by including the Szeto et al., 1999 paper.

2. A short summary of the function of all enhancers (Pen, Pit, RA4, PeIB) and the rationale for focusing on Pen1 should be considered, including their transcriptional active state during development.

We have taken this constructive suggestion as an opportunity to update the Pit enhancer by the PDE enhancer, as the PDE tested element refers to a larger transgenic construct that contains the Pit enhancer, drives expression in the mandibular arch and was extensively studied in (Sarro et al., 2018).

Updating the introduction we now write in line 64: "So far, four enhancers have been identified in mammals: PeIB which drives a distal reporter pattern in hindlimbs, PDE that drives expression in the mandibular arch, RA4 that can drive reporters in a subset of fore- and hindlimb cells and finally, Pen, a mesenchymal enhancer that drives expression in both fore- and hindlimbs (Kragesteen et al. 2018; Sarro et al. 2018; Thompson et al. 2018). Only the activity of Pen was so far shown to strongly contribute to Pitx1 function in the hindlimb as its deletion leads to a 35-50% reduction of Pitx1 expression (Kragesteen et al. 2018)."

Results

1. The authors appear to use mice in Fig 1A only for data in Fig 1B and 1C. The fact that Pitx1 is in mesenchyme is of interest but it's unclear what the purpose of these genotypes in A is. A scRNA analysis of a wt animal would have sufficed to set data shown in Fig 1B and 1C. There appears to be no other mention of the use of these three mice in the manuscript. It would be of interest to know how populations shift or change with the wt vs Pitx1^{-/-} or wt vs Pitx1D^{Pen} comparisons.

We thank the reviewer for this comment. In order to have an integrated single cell analysis, we have run from the start all the conditions together so to minimize the algorithm bias forcing the mutant dataset to fit into wildtype ones. Therefore, the analysis is from the start a combination of the 4 conditions presented in Fig1A.

According to the reviewer comment, we have now cleared this aspect in two ways:

- 1. We show in a new supplementary figure S1 (See below), the contribution of all 4 conditions (wildtype fore- and hindlimb, Pitx1^{-/-}, Pitx1^{Pen/Pen-}) to the clusters in a UMAP. We now also write in the main text line 98: "By performing unsupervised clustering of all the wildtype and mutant single cell transcriptomic datasets, we identified five clusters, to which all the dataset contributed, corresponding to the main populations of the limb: one mesenchymal cluster (Prrx1+, Prrx2+, Twist1+; 89% of the cells) and four non-mesenchymal satellite clusters including muscle (Myod1+, Ttn+; Myh3+; 4% of the cells), epithelium (Wnt6+, Krt14+; 5% of the cells), endothelium (Cdh5+, Cldn5+; 1% of the cells) and one immune cell cluster (C1qa+, Ccr1+; 1% of the cells) (Fig 1B, Fig. S1, Supplementary Table S1).*

- 2. On top of the mesenchyme cell proportion analysis, we have now quantified the proportion of each non-mesenchymal satellite clusters in our dataset and run an analysis to measure if significant changes occur in the mutant vs wildtype limbs. This analysis is displayed in a new figure S10 (see below). We have found that the proportion of non-mesenchymal satellite clusters are not affected in any of the mutant. Actually, the total proportion of mesenchyme cells itself is also not altered in any of the condition but only the proportion of mesenchymal sub-clusters, as we show in the new figure 5. We write now line 271: "First, we did not observe changes in the proportion of non-mesenchymal satellites cell clusters in any of the conditions (Fig. S10). We then measured the proportions of the different mesenchymal sub-clusters (Fig. 5A, 5B)."*

2. The hindlimb consists of 90% *Prrx1*⁺ mesenchymal cells (line 95). The 8% of *Pitx1* active cells include mesenchymal, immune, endothelium, epithelium, and muscle origin (line 134). The latter information is not well supported by the data provided in Figure 1F and seems like that the numbers do not match in the two areas, please correct.

We thank the reviewer for pinpointing these confusing quantifications. We also believe the reviewer was speaking about 8% Pitx1 inactive cells and not active ones. In the single cell analysis, we indeed define 89% of mesenchymal cells (90% in the first manuscript version) and state that Pitx1 is expressed in mesenchyme, suggesting to the reader it is not expressed in 11% of the cells. In fact, Pitx1 not solely expressed in mesenchyme and not all mesenchymal cells express Pitx1 (see updated figure S3C below). Moreover, single cell analysis alone cannot precisely define the fraction of limb cells that express Pitx1 (as only a fraction of each cell transcriptome is detected). We therefore believe that the 8% of inactive cells, defined by absence of EGFP in the cell tracing approach, is closer to the proportion of truly inactive cells. In conclusion, we believe the confusion in quantifications comes from the usage of two different methods to evaluate Pitx1 inactive cells (single cell and cell-tracing) and from the lack of precise statement of which cell types express Pitx1 or not.

To make this part of the manuscript less confusing, we now note in line 130 that some mesenchymal cells do not express Pitx1 “Yet, the enrichment of these cell types does not preclude a fraction of GFP-/Pitx1- to be of mesenchymal origin as we found a weak but clear expression of some mesenchymal markers such as Prrx1 or Twist1 in this population (Fig. S3C).”, Similarly, we can also see a few Pitx1 positive ectodermal and muscle cells in our UMAP (Fig. 1C). We noted this result in line 132: “Conversely, we found weak expression of muscle (Myh3) and ectodermal (Krt14) markers in GFP+/Pitx1+ cells (Fig. S3C).”. Finally, we now state line 104 that “Pitx1 is mostly expressed in the hindlimb mesenchymal cluster”

3. The authors did not explain the presence of the high H3K27ac activity upstream of *Pen*, close to the *H2afy* locus (Fig 3D).

This is an interesting observation. Indeed, these elements are marked with H3K27ac in GFP+ cells and not in GFP- cells, suggesting they have a relation to Pitx1 regulation. Yet, in the del1 allele, described in Kragestein and al, in 2018, that encompasses the entire region between Pitx1 to Pen, including Pen, but not these acetylated regions, only 5% of Pitx1 hindlimb expression is left. The phenotype of these animals recapitulates a full Pitx1 loss of

function. Therefore, these elements are not substantially contributing to Pitx1 hindlimb expression. We now write line 223: “We also observed a few regions upstream of Pen that were strongly enriched for H3K27ac in GFP++ cells (Fig. 3D and 3E) (Kragesteen et al. 2018). However, those sequences do not seem to be important for Pitx1 expression as the deletion of the entire region between Pitx1 and Pen, including Pen but not those regions, fully recapitulates the Pitx1 hindlimb knock out phenotype (Kragesteen et al. 2018).”

4. More outcomes after comparing Pitx1^{+/+} vs Pitx1^{GFP} and similarly Pitx1^{Pen-/Pen-} vs Pitx1^{GFP:ΔPen} transcriptomes should be included.

We agree that assessment of the effect of the EGFP transgene integration is important. Yet, as we did not have the datasets the reviewer proposes to compare, we needed to produce them. We produced RNA-seq transcriptomes of Pitx1^{+/+} and Pitx1^{GFP} somite-staged hindlimbs in biological triplicates. Pitx1 transcription remained unaltered in the EGFP sensor integrated background. Interestingly, the only observed effect was the increased transcription of a non-coding transcript directly downstream of the EGFP cassette indicating (see printscreen below). Genome-wide, we found a dozen genes differentially expressed between both conditions (adjP<0.05), however, all are either lowly expressed, non-coding or show very limited differences (log2FC<1). As this result clearly rules out a substantial effect of the EGFP cassette on Pitx1 expression, we did not produce the equivalent dataset in both Pen deleted mutants. We now write in the main result section line 116: “In order to investigate potential alterations of gene expression following the EGFP transgene integration, we produced E12.5 bulk hindlimb transcriptomes in both Pitx1^{+/+} and Pitx1^{GFP}. Here, we did not observe a change in Pitx1 expression suggesting that the insertion of the EGFP transgene did not alter Pitx1 regulation (Supplementary Table S2).”

5. Figure 4 is difficult to follow since analysis has been done in different sets of biopsies, whole hindlimb, and flow-sorted Pitx1+ hindlimb cells. In addition, the authors should explain the baseline on the expression levels in Fig4D.

Following this comment, we have now tried to make this entire part of the paper more readable by splitting figure 4 into to new figures: figure 4 and figure 5. We also did the same for the text section that is now split into two parts. Figure 4 is now dedicated to Pitx1 expression alteration in Pen-deleted hindlimbs, using both FACS cell-sorted and whole hindlimb single-cell RNA-seq. Figure 5 is now dedicated to the specific cellular alterations following the enhancer deletion. Moreover, figure S5 has now been shifted into the new figure 4E,4F.

With regards to the baseline expression levels in Fig4D we have now made it clear in the figure legend line 489: “The dotted lines indicate the threshold between low/non-expressing and intermediate-expressing (0.3).”

6. Minguillon et al 2004 propose that Pitx1 and a combinatorial code of “rostral” Hox genes trigger the Tbx4 expression. The authors should determine the Hox cluster alterations in the absence of Pen1.

We find this an interesting suggestion. We investigated differential gene expression between Pitx1^{Pen-/Pen-} and wildtype in the mesenchyme and all of its sub-clusters. This analysis is now presented in a new Table S5. In some cell clusters (EDC, DP, DPP, TP, ICT), we found a concomitant loss of some Hox genes and Tbx4, however, in other cluster the loss of Hox genes was not associated to a loss of Tbx4 (Ms, LDC, PC) and finally in the proximal proliferative progenitor cluster, where Tbx4 shows the strongest expression decrease, we find no alteration in Hox gene expression. This suggests that the loss of Tbx4 expression is not linked to Hox gene but is rather a direct downstream effect of Pitx1 expression loss. We write line 306: “To further determine the origin of the Tbx4 loss we assessed in Pitx1^{Pen-/Pen-} hindlimb clusters the expression of caudal Hox genes, which have been suggested to control Tbx4 along with Pitx1 (Minguillon et al. 2005). Here, we did not find an alteration in Hox expression levels that correlates with Tbx4 loss, suggesting that Tbx4 decrease is rather a direct effect of Pitx1 loss-of-expression (Supplementary Table S5).”

Discussion

1. The potential function of the other enhancers should be included.

We appreciate this comment and now write in the discussion in line 401: "Therefore, we hypothesize that the role of Pen is not to act as a pattern-defining enhancer but rather as a support enhancer that ensures a robust transition of cells towards a fully active Pitx1 landscape and therefore a strong Pitx1 transcription. Here, other enhancers, such as RA4, PeIB as well as other to be defined enhancers, might bear this pattern-defining role."

2. The developmental/evolutionary conservation of the Pen1 enhancer should be included.

We now mention the evolutionary and developmental conservation of Pen in the discussion line 386: "Here we have tested how the loss of one of the regulatory elements, the Pen enhancer, which is conserved among all tetrapods and required for hindlimb identity, affects the establishment of the Pitx1 active landscape (Domyan et al. 2016; Kragestein et al. 2018)."

Minor points

A brief description of the supplementary tables would facilitate the data interpretation.

We have added such description on the last page of the supplementary info.

Reviewer #3 (Remarks to the Author):

(I) Summary/Findings

The authors combine both genetic mouse models and high throughput sequencing techniques to outline the role of the Pitx1 enhancer and the local 3D chromatin architecture in hindlimb development. The authors provide evidence to show that indeed the Pitx1 regulatory locus functions as an entire unit and is not uniquely and fully dependent on the Pen enhancer. This data is supported by their H3K27ac and 3D chromatin conformation assays.

These results nicely outline the effects of loss of a known Pitx1 enhancer. The authors find that loss of Pen leads to an increase in the number progenitor cells as evidenced by their scRNA-seq data at the expense of proximal, Pitx1 high expressing cells. Using their newly generated mouse model that tracks the expression of Pitx1 via GFP, the authors are able to confirm that indeed loss of Pen leads to reduced Pitx1 expression (as tracked by GFP), leading to an increase in the proportion of GFP-lower expressing cells that were shown to exhibit H3K27ac at known distal genes. Interestingly, loss of Pitx1 leads to an increase in progenitor cells (so fewer proximal and distal differentiated cells), while loss of Pen leads to a similar phenotype but only in the proximal cell clusters, highlighting the role of the Pen enhancer in proximal cells. The increase in progenitor cells observed in the Pitx1^{-/-} and Pen^{-/-} mice may thus contribute to the clubfoot phenotype, which may be the result of impaired differentiation of cells that contribute to hindlimb structures.

Overall, the authors characterize the role of the Pen enhancer during limb development. Indeed, this is important to further our understanding of enhancer functions, specifically when establishing both gene expression and chromatin landscapes during development and thus would provide significant knowledge to the field.

We thank the reviewer for their positive appreciation of our work as well as for all the constructive suggestions made.

(II) Major comments

1. The authors suggest that the differences in Pitx1 expression is associated with a progressive increase in its cis-regulatory landscape rather than the activity of individual enhancers. What might modulate the expression of Pitx1 expression? Could there be other chromatin conformations that are not at the analyzed locus that could modulate the expression of Pitx1?

We thank this reviewer for these questions. Given the fact that the expression seems to vary among the proximo-distal axis we tend to think that proximally-expressed genes or proximo-distal signaling cues might control the activity of the landscape. These factors could then influence the landscape activity by altering, for instance, the proximity to repressive nuclear lamina. Alternatively, it is possible that small increase/decrease in chromatin interactions within the active locus 3D structure contributes to these changes. We now write in the discussion line 382: "This suggests that proximal transcription factors or signaling cues are controlling the landscape either by binding simultaneously at several Pitx1 cis-regulatory regions or by targetedly modifying other parameters of locus activity such as the frequencies of active chromatin interactions or the proximity to the repressive nuclear lamina for instance."

2. Besides Tbx4, are there other genes that Pitx1 regulates whose expression is altered that might lead to the increased numbers of distal cells? Can the authors look into the expression of Pitx1-regulated genes to try and link the observed phenotype with the bimodal levels of Pitx1?

We appreciate this comment; however, we disagree with one of his points: the number of distal cells is not increased in Pitx1^{Pen-/Pen-} or in Pitx1^{-/-}. In fact, the clusters that show (Fig. 5B) the most increased in number of cells are proximal proliferative progenitors (PPP) in Pitx1^{Pen-/Pen-} and proximal and distal proliferative progenitors (PPP and DPP) in Pitx1^{-/-} (Fig. 5B). Therefore, our hypothesis is rather that alteration of Pitx1 expression leads to a poorer differentiation of proximal progenitor cells in Pitx1^{Pen-/Pen-}, which show a higher number of non/low-Pitx1 expressing cells in mutants (Fig. 4D).

However, as suggested by the reviewer, we also looked in more detail into differentially expressed genes between wildtype and Pitx1^{Pen-/Pen-} hindlimbs that could explain this possible differentiation delay. First, we observed an increase of progenitors' marker genes like Hist1h1 genes supporting the increased pool of proximal proliferative progenitors (PPP). Yet, we could not formally link changes in cell proportions to differentially regulated genes although we saw that important limb patterning genes are downregulated like Twist1 or Hoxd11, which knock out lead to zeugopod-digits phenotypes as well. We also found de-regulation of distal genes like Hoxa13 or Irx1, which are marking the delay in development of the limb rather than causing the proximal effects. We also observe that Dcn, an important gene for tendons mechanical properties that is express in ICT is partially decreased. Six1 and Six2 genes, that are necessary for muscle formation were also found dysregulated in the mutants. We now provide a new list of differential gene expression in the mesenchyme and all of its sub-clusters (Table S5) that is referred to in the main text line 315: "Moreover, aside from Tbx4, we found numerous dysregulated genes in Pitx1^{Pen-/Pen-}

hindlimbs which might contribute to observed phenotypes (Supplementary Table S5). This is the case for *Dcn*, which was previously described to be involved in tendon elasticity in mice as well as the *Six1* and *Six2* genes that are expressed in connective tissue and necessary for skeletal muscle development (Bonnin et al. 2005; Dourte et al. 2012; Wu et al. 2014; Robinson et al. 2017)."

As suggested by the reviewer, in order to link changes in gene expression and the bimodal expression of *Pitx1*, we have looked into known *Pitx1*-dependant genes from Nemeč et al., 2017 and compared them with a newly generated GFP^{-/-} and GFP^{+/+} RNA-seq dataset. In 50% of the cases, the expression of *Pitx1*-dependant gene was significantly different in GFP^{-/-} and GFP^{+/+} and in most of the cases, the direction of the difference was the same as the one of *Pitx1*. This is now represented in a new panel in figure S7 and differentially expressed genes between these two populations can be found in Supplementary Table S4. We have now written a description of these in the manuscript line 210: "On average, we found three times more *Pitx1* transcripts in GFP^{+/+} cells than in GFP^{-/-} cells as well as an enrichment for several known *Pitx1* target genes including *Tbx4* (Fig. 3B, Fig. S7A, S7B, Supplementary Table S4) (Nemeč et al. 2017)."

3. Are there any other loci involved in proximal limb development that show altered H3K27ac in the GFP^{-/-} vs GFP^{+/+} (or in the Pen^{-/-} mice)? As the authors remark, a failure to lead to some threshold of gene expression may lead to the observed clubfoot phenotype, but more analysis should be added to support this claim.

Linking this question to the previous (question 2), we have produced, on top of the H3K27ac dataset, new RNA-seq dataset from intermediated- (GFP^{-/-}) and high- (GFP^{+/+}) hindlimb wildtype expressing cells to map potentially affected loci displaying differential activity. The transcriptomic data further support that the expression of *Pitx1* follows the GFP activity and that each cell population has a distinct identity. Non/low-expressing cells show enrichment for satellite cluster markers. Intermediate-expressing cells are enriched for distal markers. Finally, high expressing cells are enriched for proximal and irregular connective tissue markers. We also show that this translates into differences in H3K27ac. We have now integrated this new data in figure 3 and shifted part of figure 3 to figure S7. Moreover, we have produced a new supplementary table S4 showing differentially expressed genes between the 3 cell populations.

We now write in the text line 210: "On average, we found three times more *Pitx1* transcripts in GFP^{+/+} cells than in GFP^{-/-} cells as well as an enrichment for several known *Pitx1* target genes including *Tbx4* (Fig. 3B, Fig. S7A, S7B, Supplementary Table S4) (Nemeč et al. 2017). Moreover, as expected from the single-cell analysis, high-

expressing GFP⁺⁺ cells were mostly enriched for proximal limbs markers (*Shox2*, *Gsc*, *Tbx18*, *Tbx3* and *Hoxa11*) and showed higher expression of ICT marker genes (*Kera* and *Lum*) (Fig. 3C, and Fig. S7C, Supplementary Table S4). In contrast, intermediate-expressing cells GFP^{+/-} were enriched for distal cell markers (*Hoxa13*, *Hoxd13*, *Wnt5a*, *Lhx2*, *Msx1*) (Fig. 3C, Fig. S7C, Supplementary Table S4).”

4. Authors suggest that the clubfoot phenotype may be the result of increased progenitor cells, however this claim is not clearly supported. Can the authors identify more targets (such as *Tbx4* or others) and use an in vitro model to evaluate whether restoration of these targets is able to support differentiation of a proximal cell phenotype. Or are the progenitor cells in the *Pen*^{-/-} mice more poorly differentiated at later stages of development? Perhaps a time-course experiment (ideally with scRNA-seq data if feasible) would help to better link these data with the clubfoot phenotype.

These are compelling suggestions that we have answered in several ways:

- 1- In a new differential expression analysis between wildtype and *Pen*^{-/-}, we see a significant increase of progenitor markers. We now write line 283: “The increase in PPP cells is further supported by the upregulation of its markers in *Pitx1*^{*Pen*^{-/-}} hindlimbs (*Hist1h* genes, *Top2a* and others; Supplementary Table S5).”
- 2- We also refined our hypothesis regarding cell populations using velocity analysis of proximal and distal limb clusters that is now displayed in figure 5C and 5D (see below). This enabled us to support our hypothesis of a delay in PPP to ICT differentiation at E12.5. As we explain at line 286; “To test if these effects could be explained by a delayed differentiation of progenitor cells, we performed a velocity analysis on *Pitx1*^{-/-}, *Pitx1*^{*Pen*^{-/-}} and *Pitx1*^{+/+} limbs in proximal and distal cell clusters separately. In the proximal part of the hindlimb, we found in both mutants a predicted connection from PPP to ICT cells, suggesting an ongoing differentiation process (Fig. 5C). This connection was not present in *Pitx1*^{+/+} fore- and hindlimbs, suggesting that the differentiation process was completed in these tissues. These findings are further supported by an increase of ICT marker genes (*Lum*, *Dcn* and *Kera*) in PPP cells of both mutants, suggesting that those cells have only partially adopted an ICT identity but still did not fully differentiate (Fig. S11).”

3- With regard to effect of the alteration of the PPP to ICT ratio at later developmental stages, we believe that a comprehensive analysis of late limb development and, in particular, of musculo-skeletal patterning would be outside of the scope of this manuscript. Yet, we have now carefully stated our hypothesis for the mechanism leading to clubfoot, as well as an alternative one, in the discussion on line 421 “Here, the particularly strong effect of the *Pen* deletion on the ICT cell proportion pinpoints these cells as the origin of the clubfoot phenotype seen in mice lacking the enhancer. In fact, ICT has been repeatedly reported to function in a non-cell autonomous way during limb development and to act as an important driver of muscle patterning (Chevallier and Kieny 1982; Kardon et al. 2002; Kardon et al. 2003; Swinehart et al. 2013; Colasanto et al. 2016; Vallecillo-Garcia et al. 2017; Besse et al. 2020). We therefore suspect that the loss of ICT in hindlimbs leads to a muscle patterning defect which would be at the base of the clubfoot phenotype. An alternative hypothesis could be that the observed heterochrony in several of the mesenchymal cell populations could collectively cause the clubfoot since coordinated expansion and interactions among different mesenchymal cell populations are required for normal limb morphogenesis.”. Of note to this reviewer, we believe that the reduction of ICT tissue proportion indeed results in a muscle patterning defect as we have preliminary data showing defects in muscle bundle separation (see the *Myog* In Situ Hybridisation below, altered muscle bundles are shown by a white arrow). Yet, as we believe a full characterization of this phenotype would require more evidence and would be outside the scope of this manuscript, we prefer not to include this piece of data in the manuscript.

5. The authors highlight reduced number of *Tbx4* expressing cells in *Pen*^{-/-} mice being similar to loss of *Pitx1*^{-/-} and suggest that loss of *Pitx1* expression by deletion of the enhancer might be mediated by *Tbx4*. Are the *Tbx4*-negative cells that result as a loss of *Pen* or *Pitx1* transcriptionally similar? Can the authors show a UMAP plot of the scRNA-seq data to highlight the changes in the proportions of cells (to better compare and visualize the difference or similarities between the two genotypes)?

We thank the reviewer for this comment. We think that our hypothesis was not stated clearly. Indeed, we think that the loss of the enhancer has a direct cis-effect on *Pitx1* expression and not a *Tbx4*-mediated one. In fact, the deletion of *Pen* induces a loss of *Pitx1* and the loss of *Pitx1* affects *Tbx4* expression. We now produced a better comparison between *Tbx4* expressing and non-expressing cells in *Pitx1*^{-/-}, *Pitx1*^{Pen-/Pen-} and hindlimb wildtype in each mesenchymal sub-cluster in figure S12 and we re-wrote the entire section about the loss of *Pitx1* and the subsequent loss of *Tbx4* line 300: “As *Pitx1* has been shown to have both indirect and direct downstream effects, we further investigated differentially expressed genes in *Pitx1* loss-of-function hindlimbs that could induce these effects. In particular it has been shown that *Tbx4*, a known downstream target gene of *Pitx1*, mediates the *Pitx1*-effect on hindlimb buds growth rate (Oumette et al. 2010; Duboc and Logan 2011; Nemeč et al. 2017). As anticipated, we found a downregulation of the *Tbx4* in all clusters aside of PC, Ms and LDC in both *Pitx1*^{-/-} and *Pitx1*^{Pen-/Pen-} hindlimbs, with the strongest effect in ICT and PPP clusters (Fig. S12A-D, Supplementary Table S5).” Finally, we have also included, on top of the one included in figure 5A, UMAPs of the 4 dataset and plotted the expression of key marker genes in the figure S11 (see below) to highlight the difference of cell proportions. We write line 297: “Together, these findings support a form of heterochrony that affects only the proximal part of *Pitx1*^{Pen-/Pen-} hindlimbs and that is featured by a delayed differentiation of PPP to ICT.”

6. Did the authors detect any interchromosomal contacts or other long-range contacts that might be altered in the *Pitx1*^{-/-} and *Pen*^{-/-} models?

*This is an interesting thought as indeed partial loss of function and full loss of function of transcription factors might influence the genomic architecture of the cells. In this project, we have not produced HiC data as this goes above the scope of this research. Yet, we have now computed a full chromosome 13 interaction map and did not see changes of long-range contact from the *Pitx1* promoter between wt and *Pitx1*^{Pen-/Pen-} embryo both in GFP⁺ and GFP⁻ cell populations. We included this in the new figure S14D (see below) that is referred to at line 357.*

7. The cells that are able to maintain the GFP+ expression under loss of Pen maintain similar RNA profiles; which suggests there may also be other factors that contribute to the observed local chromatin conformation that supports active Pitx1 transcription. Can the authors speculate on what some of these factors might be?

As the cell identity of GFP+ cells do not change between wildtype and Pitx1^{Pen-/Pen-}, we think that the change is not due to the presence of different factors. The robustness in activity of the cis-regulatory landscape is given by the presence of the activation co-factors that bind to the various Pitx1 enhancers. The loss of Pen could simply decrease this robustness. Alternatively, non-transcriptional differences in cell identity could enable those cells to resist the loss of Pen. To include these suggestions we write line 388: "Some escaping cells can induce Pitx1 regulatory landscape activation without Pen, suggesting that the other cis-regulatory modules (PeIB, PDE and RA4) provide a form a compensation. These modules are likely activated by a similar gene-regulatory network in wildtype and mutant hindlimbs, as we could not observe a clear shift in cell identity of GFP/Pitx1 expressing cells. Alternatively, a cumulative effect of marginal transcriptional changes in cell identity along with specific non-transcriptional identity differences could maintain the capacity of cells to generate a high Pitx1 expression level despite the absence of Pen."

(III) Minor comments

1. Figure 1B: Should include UMAP colored by sample/genotype.

We appreciate this comment and as it is difficult in a single UMAP to keep both the cluster information (cell type) and the origin of sample (especially for 4 samples), we have produced this latter information in a supplementary figure S1 (see below), that shows how all samples (wildtype fore- and hindlimb, Pitx1^{-/-}, Pitx1^{Pen-/Pen-}) contribute to all the clusters. This figure is referred to in the main text line 103.

2. Authors state that in GFP- cells deleted for Pen, there was ectopic acetylation of Pitx1; can the authors look at the expression of GFP as well to see if its significantly enriched in this cluster to further support their claim that this is due to the relocation of GFP- cells in the gating strategy.

The reviewer is correct, we believe that GFP- cells enriched for acetylation are cells that would normally express EGFP and Pitx1 strongly and that are relocated in the GFP- gating due to the loss of the active regulatory landscape. The strongest evidence for the relocation of these cells, is the strong increase of mesenchymal marker genes (Figure 6A), normally associated to EGFP and Pitx1 expression in wildtype hindlimb, in the GFP-/Pitx1-fraction of Pen-deleted hindlimbs. As suggested by the reviewer we have now measured EGFP expression in GFP-cells and found no significant expression difference between Pen-deleted and wildtype cells. This new analysis is presented in figure S8C (see below). To integrate these results in our manuscript we now write line 240: "To confirm that this effect is not due to a difference in the distribution of EGFP fluorescence during cell sorting, we compared EGFP transcription in Pitx1^{GFP;ΔPen} and Pitx1^{GFP} GFP- cells and did not observe a difference (Fig. S8C). ". Interestingly, the absence of GFP expression change differs from the very mild expression increase we see for Pitx1 in these cells. We interpret this result by the fact that in the relocated GFP- cells, the locus is not fully repressed but is in-between repression and activation, which is likely linked to a slightly higher leakage from Pitx1 promoter. We write this at line 345: "These activities are likely caused by the relocation, in the GFP- fraction, of cells that would normally express Pitx1 but fail to establish a fully active landscape in the absence of Pen. In these cells, we observed a marginal increase in Pitx1 expression (FC=1.6, padj=0.0026) that suggests that the locus is less repressed as in wildtype GFP- cells (Fig. 6C, Supplementary Table S6). "

3. Generally, more UMAP plots need to be added to show the differences in the cell type distribution across the various genotypes.

We have added UMAPs on supplementary figures S1, S6, S11 and S12 to improve the understanding of changes in expression and cell population.

Reviewer #4 (Remarks to the Author):

This manuscript reports the characterization of Pitx1 regulatory regions in sorted cells derived from the mouse limb. Deletion of the Pen enhancer (Pitx1 Pen⁻/Pen⁻) resulted in decreased number of cells expressing high Pitx1 levels and increased cells expressing low Pitx1 levels. scRNA-seq allowed the identification of several cell clusters, expressing different Pitx1 levels. In cells not expressing Pitx1, the Pen enhancer contacted the repressed Neurog1 gene. Pitx1 Pen⁻/Pen⁻ cells also displayed reduced Tbx4 expression. Some Pitx1 Pen⁻/Pen⁻ cells (high-expressing escaping cells) expressed high Pitx1 level. H3K27 acetylation landscape was indistinguishable in WT GFP⁺ (high expressing cells) and Pitx1 Pen⁻/Pen⁻ (high-expressing escaping cells). 3D conformation of Pitx1 Pen⁻/Pen⁻ (high-expressing escaping cells) was different from that adopted by WT cells, indicating that escaping cells do not require Pen to establish an active 3D conformation. Slight reduction of contact between the Pitx1 promoter and another enhancer (Pit/RA4) was noted.

The experiments are well conducted and controlled and the findings of interest. There are few issues that requiring clarification.

We thank the reviewer for their appreciation of our work as well as for all the constructive suggestions made.

1. Figure 1G. A higher-definition map would allow better visualization of the contacts established by Pitx1 and its regulatory elements with Neurog1.

We have now produced high-resolution 1kb maps with KR normalisation (Fig. S5, see below) and aligned them with repressive and active histone marks maps. We have highlighted the main contacts to show how they match the specific histone modifications domains. However, because of the sparsity of the signal, it not possible to subtract the two different 1kbp maps, we therefore have left the 5kb-resolution maps in the main figure. We now refer to this figure at lines 143 and 147.

2. Figure S7. How does the Pen deletion affect Tbx4? Is this a direct or indirect effect?

The reviewer makes an important point that we have now made clearer: we believe, in agreement with previous publications in the field that this effect is produced indirectly through the loss of Pitx1 expression, a known regulator of Tbx4 (Duboc and Logan, 2011, Quimette et al., 2010 and Nemeč et al., 2017). In the revised version we also provide a differential gene expression analysis between Pitx1^{+/+} and Pitx1^{Pen⁻/Pen⁻} in supplementary table S5 for each mesenchymal subcluster that shows that the loss of Tbx4 expression and Tbx4-expressing cell proportion coincides with Pitx1 loss. We have now clearly stated the indirect link between Pen deletion and Tbx4 in the text line 302 "In particular it has been shown that Tbx4, a known downstream target gene of Pitx1, mediates the Pitx1-effect on hindlimb buds growth rate (Quimette et al. 2010; Duboc and Logan 2011; Nemeč et al. 2017)" and line 417: "Although the reduction of Pitx1 transcription induces the decrease of its direct target, Tbx4..."

3. While Pen enhancer is clearly important to regulate Pitx1 expression, the lack of 3D active contacts and similar H3K27ac landscape in Pitx1 Pen-/Pen- high-expressing escaping cells suggest its dispensability. Perhaps subpopulation(s) of mesenchymal cells, not identified in this study, employ a distinct regulatory network to regulate Pitx1 expression. The authors may want to speculate on this point.

As the cell identity of GFP+ cells do not change between wildtype and Pitx1^{Pen-/Pen-}, we indeed think that the change is not due to the presence of different factors. The robustness in activity of the cis-regulatory landscape is given by the presence of factors that bind to the various Pitx1 enhancers. The loss of Pen could simply decrease this robustness. Alternatively, non-transcriptional differences in cell identity could enable those cells to resist the loss of Pen. To make this point clearer we now write line 388: "Some escaping cells can induce Pitx1 regulatory landscape activation without Pen, suggesting that the other cis-regulatory modules (PeIB, PDE and RA4) provide a form a compensation. These modules are likely activated by a similar gene-regulatory network in wildtype and mutant hindlimbs, as we could not observe a clear shift in cell identity of GFP/Pitx1 expressing cells. Alternatively, a cumulative effect of marginal transcriptional changes in cell identity along with specific non-transcriptional identity differences could maintain the capacity of cells to generate a high Pitx1 expression level despite the absence of Pen."

Reviewers' Comments:

Reviewer #1:

Remarks to the Author:

Rouco and coauthors have undertaken an excellent effort to address all criticisms by the reviewers and included the necessary additional data and revised the text accordingly. All issues raised by this reviewer in the first review have been addressed in an overall satisfactory manner. Only minor comments remain that the authors should correct as this will improve the manuscript further. In this reviewer's opinion, the manuscript can be accepted after dealing with these minor revisions. This study provides important novel and functional insights of general relevance into the cell-specific alterations and compensatory mechanism following germline deletion of a single enhancer that participates in Pitx1 expression in hindlimb buds.

Minor comments

Line 317: decorin (Dcn) is also a connective tissue marker in limb buds – this should be mentioned in the text.

Line 326: the sentence starting with: In GFP+ and GFP- is and missing information.
Suggestion: In GFP+ and GFP- cells from Pitx1GDF;dPen hindlimb buds, we used RNA-seq to assess whether..... loss as observed using scRNA-seq.

Line 337: We then..... and in the increased fraction of GFP- cells.

Lines 362-364: The conclusion lacks some important information for understanding. Suggestion: This shows that despite some remaining regulatory activity (evidenced by low level H3K27ac; arrows Fig. 6C), the locus is unable.... transcribe Pitx1 (Fig. 6G)

Lines 421-430: rather than postulating two alternate hypotheses – is it not equally likely that both muscle patterning and altered expansion and interactions among mesenchymal populations cause clubfoot as limb muscle and skeletal development are linked?

On a speculative notion: is the molecular mechanism that allows a fraction of cells to “escape” some sort of adaptation of the entire cis-regulatory landscape in these cells due to the fact that the Pen enhancer is deleted from the germline. This is a curiosity driven comment rather than an issue to be addressed.

Reviewer #2:

Remarks to the Author:

The study by Rouco et al addresses the cell-type specificity during hindlimb development of one of the Pitx1 enhancers. It combines mouse molecular genetics and computational analyses. The authors responded to the reviewer's comments and suggestions. The data are convincing and support the conclusions and claims. Data analysis and interpretation are well stated and the technology is state-of-the-art. The work is significant to the field of developmental biology.
Chrissa Kioussi

Reviewer #3:

Remarks to the Author:

Summary/Findings:

The authors combine both genetic mouse models and high throughput sequencing techniques to outline the role of the Pitx1 enhancer and the local 3D chromatin architecture in hindlimb development. The authors provide evidence to show that indeed the Pitx1 regulatory locus functions as an entire unit and is not uniquely and fully dependent on the Pen enhancer. This data is supported by their H3K27ac and 3D chromatin conformation assays. These results nicely outline the effects of loss of a known Pitx1 enhancer. The authors find that loss of Pen leads to an increase in the number progenitor cells as evidenced by their scRNA-seq data at the expense of

proximal, Pitx1 high expressing cells. Using their newly generated mouse model that tracks the expression of Pitx1 via GFP, the authors are able to confirm that indeed loss of Pen leads to reduced Pitx1 expression (as tracked by GFP), leading to an increase in the proportion of GFP-lower expressing cells that were shown to exhibit H3K27ac at known distal genes. Interestingly, loss of Pitx1 leads to an increase in progenitor cells (so fewer proximal and distal differentiated cells), while loss of Pen leads to a similar phenotype but only in the proximal cell clusters, highlighting the role of the Pen enhancer in proximal cells. The increase in progenitor cells observed in the Pitx1-/- and Pen-/- mice may thus contribute to the clubfoot phenotype, which may be the result of impaired differentiation of cells that contribute to hindlimb structures. Overall, the authors characterize the role of the Pen enhancer during limb development. Indeed, this is important to further our understanding of enhancer functions, specifically when establishing both gene expression

The authors have also substantially improved the manuscript in response to the review comments and have provided sufficient evidence to substantiate the claims that were previously not as clear.

Specific responses to major comments in the original review:

1. The authors suggest that the differences in Pitx1 expression is associated with a progressive increase in its cis-regulatory landscape rather than the activity of individual enhancers. What might modulate the expression of Pitx1 expression? Could there be other chromatin conformations that are not at the analyzed locus that could modulate the expression of Pitx1?
 - a. The authors have added to the discussion that this may be a point to consider, which is a satisfactory response.
2. Besides Tbx4, are there other genes that Pitx1 regulates whose expression is altered that might lead to the increased numbers of distal cells? Can the authors look into the expression of Pitx1-regulated genes to try and link the observed phenotype with the bimodal levels of Pitx1?
 - a. The authors clarify a misunderstanding.
 - b. Additionally, the authors nicely include additional analysis of other genes whose expression is dysregulated in Pitx1-Pen-/- mice that might also contribute to the observed phenotype. The inclusion of more RNA-seq data on GFP+- and GFP++ cells also further supports the role of this enhancer in regulating the expression of Pitx1 and thus subsequent targets.
3. Are there any other loci involved in proximal limb development that show altered H3K27ac in the GFP+- vs GFP++ (or in the Pen-/- mice)? As the authors remark, a failure to lead to some threshold of gene expression may lead to the observed clubfoot phenotype, but more analysis should be added to support this claim
 - a. The authors address of this concern is satisfactory.
4. Authors suggest that the clubfoot phenotype may be the result of increased progenitor cells, however this claim is not clearly supported. Can the authors identify more targets (such as Tbx4 or others) and use an in vitro model to evaluate whether restoration of these targets is able to support differentiation of a proximal cell phenotype. Or are the progenitor cells in the Pen-/- mice more poorly differentiated at later stages of development? Perhaps a time-course experiment (ideally with scRNA-seq data if feasible) would help to better link these data with the clubfoot phenotype.
 - a. Inclusion of the RNA velocity analysis is satisfactory to address this concern, and indeed perhaps some of the additional suggested experiments would be beyond the scope of this study.
5. The authors highlight reduced number of Tbx4 expressing cells in Pen-/- mice being similar to loss of Pitx1-/- and suggest that loss of Pitx1 expression by deletion of the enhancer might be mediated by Tbx4. Are the Tbx4- negative cells that result as a loss of Pen or Pitx1 transcriptionally similar? Can the authors show a UMAP plot of the scRNA-seq data to highlight the changes in the proportions of cells (to better compare and visualize the difference or similarities between the two genotypes)?
 - a. The authors' response to this comment is satisfactory.
6. Did the authors detect any interchromosomal contacts or other long-range contacts that might be altered in the Pitx1-/- and Pen-/- models?
 - a. The authors include full chromosome 13 analysis, which is adequate to address this comment.
7. The cells that are able to maintain the GFP+ expression under loss of Pen maintain similar RNA profiles; which suggests there may also be other factors that contribute to the observed local chromatin conformation that supports active Pitx1 transcription. Can the authors speculate on what some of these factors might be?

a. The authors have included additional comments in the discussion to address this point, which is satisfactory.

Overall, the manuscript is substantially improved with suggestions provided by all reviewers.

Reviewer #4:

Remarks to the Author:

The authors have satisfactorily addressed my comments.

REVIEWERS' COMMENTS

Reviewer #1 (Remarks to the Author):

Rouco and coauthors have undertaken an excellent effort to address all criticisms by the reviewers and included the necessary additional data and revised the text accordingly. All issues raised by this reviewer in the first review have been addressed in an overall satisfactory manner. Only minor comments remain that the authors should correct as this will improve the manuscript further. In this reviewer's opinion, the manuscript can be accepted after dealing with these minor revisions. This study provides important novel and functional insights of general relevance into the cell-specific alterations and compensatory mechanism following germline deletion of a single enhancer that participates in Pitx1 expression in hindlimb buds.

We thank this reviewer for their positive assessment of our answer to their comments and suggestions. We have now integrated the suggested changes in the manuscript.

Minor comments

Line 317: decorin (Dcn) is also a connective tissue marker in limb buds – this should be mentioned in the text. *This is correct, and as we use Dcn as a marker gene for irregular connective tissue (ICT) (as defined by Besse et al., 2020) across the manuscript. We now write line 307: “This is the case for Dcn, an ICT marker gene previously described to be involved...”*

Line 326: the sentence starting with: In GFP+ and GFP- is and missing information. Suggestion: In GFP+ and GFP- cells from Pitx1^{GDF;dPen} hindlimb buds, we used RNA-seq to assess whether..... loss as observed using scRNA-seq.

We have now corrected the sentence and wrote line 316: “In GFP+ and GFP- cells from Pitx1^{GFP;ΔPen} hindlimb buds, we used RNA-seq to assess whether we could observe similar changes in cellular identity upon Pen enhancer loss as the one previously described using scRNA-seq.”

Line 337: We then..... and in the increased fraction of GFP- cells.

We have now corrected the sentence line 327: “We then performed H3K27ac ChIP-seq in the escaping GFP+ cells and in the increased fraction of GFP- cells.”

Lines 362-364: The conclusion lacks some important information for understanding. Suggestion: This shows that despite some remaining regulatory activity (evidenced by low level H3K27ac; arrows Fig. 6C), the locus is unable.... transcribe Pitx1 (Fig. 6G)

We have now changed the sentence line 352: “This shows that despite some remaining regulatory activity (evidenced by low level H3K27ac; arrows Fig. 6C), the locus is unable to assume its active 3D structure and therefore to efficiently transcribe Pitx1 (Fig. 6G).”

Lines 421-430: rather than postulating two alternate hypotheses – is it not equally likely that both muscle patterning and altered expansion and interactions among mesenchymal populations cause clubfoot as limb muscle and skeletal development are linked?

We agree with this reviewer, and in order to avoid making alternate hypothesis, we just started the sentence (now line 415) by “moreover” instead of “an alternative hypothesis”.

On a speculative notion: is the molecular mechanism that allows a fraction of cells to “escape” some sort of adaptation of the entire cis-regulatory landscape in these cells due to the fact that the Pen enhancer is deleted from the germline. This is a curiosity driven comment rather than an issue to be addressed.

This is an interesting idea. I would rather believe that the effect happens only when the landscape activates itself during hindlimb development, therefore in a non-instructive way. At Pitx1, it is likely to be the case, because the entire locus changes extensively its structure and E-P interactions patterns in active tissue and thus the loss of one of the involved elements will have an effect solely when this structure is changing (and here adapting to this loss). As such, it would predict that a limb-restricted deletion of Pen would induce the same effect. Yet, this remains to be shown and might also be very different at other loci, especially the ones with rather stable, instructive interaction patterns.

Reviewer #2 (Remarks to the Author):

The study by Rouco et al addresses the cell-type specificity during hindlimb development of one of the Pitx1 enhancers. It combines mouse molecular genetics and computational analyses. The authors responded to the reviewer's comments and suggestions. The data are convincing and support the conclusions and claims. Data analysis and interpretation are well stated and the technology is state-of-the-art. The work is significant to the

field of developmental biology.
Chrissa Kioussi

We thank Prof. Kioussi for her positive assessment of our manuscript and answer to her comments and suggestions.

Reviewer #3 (Remarks to the Author):

Summary/Findings:

The authors combine both genetic mouse models and high throughput sequencing techniques to outline the role of the Pitx1 enhancer and the local 3D chromatin architecture in hindlimb development. The authors provide evidence to show that indeed the Pitx1 regulatory locus functions as an entire unit and is not uniquely and fully dependent on the Pen enhancer. This data is supported by their H3K27ac and 3D chromatin conformation assays. These results nicely outline the effects of loss of a known Pitx1 enhancer. The authors find that loss of Pen leads to an increase in the number progenitor cells as evidenced by their scRNA-seq data at the expense of proximal, Pitx1 high expressing cells. Using their newly generated mouse model that tracks the expression of Pitx1 via GFP, the authors are able to confirm that indeed loss of Pen leads to reduced Pitx1 expression (as tracked by GFP), leading to an increase in the proportion of GFP-lower expressing cells that were shown to exhibit H3K27ac at known distal genes. Interestingly, loss of Pitx1 leads to an increase in in progenitor cells (so fewer proximal and distal differentiated cells), while loss of Pen leads to a similar phenotype but only in the proximal cell clusters, highlighting the role of the Pen enhancer in proximal cells. The increase in progenitor cells observed in the Pitx1^{-/-} and Pen^{-/-} mice may thus contribute to the clubfoot phenotype, which may be the result of impaired differentiation of cells that contribute to hindlimb structures. Overall, the authors characterize the role of the Pen enhancer during limb development. Indeed, this is important to further our understanding of enhancer functions, specifically when establishing both gene expression

The authors have also substantially improved the manuscript in response to the review comments and have provided sufficient evidence to substantiate the claims that were previously not as clear.

Specific responses to major comments in the original review:

1. The authors suggest that the differences in Pitx1 expression is associated with a progressive increase in its cis-regulatory landscape rather than the activity of individual enhancers. What might modulate the expression of Pitx1 expression? Could there be other chromatin conformations that are not at the analyzed locus that could modulate the expression of Pitx1?
 - a. The authors have added to the discussion that this may be a point to consider, which is a satisfactory response.
2. Besides Tbx4, are there other genes that Pitx1 regulates whose expression is altered that might lead to the increased numbers of distal cells? Can the authors look into the expression of Pitx1-regulated genes to try and link the observed phenotype with the bimodal levels of Pitx1?
 - a. The authors clarify a misunderstanding.
 - b. Additionally, the authors nicely include additional analysis of other genes whose expression is dysregulated in Pitx1-Pen^{-/-} mice that might also contribute to the observed phenotype. The inclusion of more RNA-seq data on GFP⁺ and GFP⁺⁺ cells also further supports the role of this enhancer in regulating the expression of Pitx1 and thus subsequent targets.
3. Are there any other loci involved in proximal limb development that show altered H3K27ac in the GFP⁺ vs GFP⁺⁺ (or in the Pen^{-/-} mice)? As the authors remark, a failure to lead to some threshold of gene expression may lead to the observed clubfoot phenotype, but more analysis should be added to support this claim
 - a. The authors address of this concern is satisfactory.
4. Authors suggest that the clubfoot phenotype may be the result of increased progenitor cells, however this claim is not clearly supported. Can the authors identify more targets (such as Tbx4 or others) and use an in vitro model to evaluate whether restoration of these targets is able to support differentiation of a proximal cell phenotype. Or are the progenitor cells in the Pen^{-/-} mice more poorly differentiated at later stages of development? Perhaps a time-course experiment (ideally with scRNA-seq data if feasible) would help to better link these data with the clubfoot phenotype.
 - a. Inclusion of the RNA velocity analysis is satisfactory to address this concern, and indeed perhaps some of the additional suggested experiments would be beyond the scope of this study.
5. The authors highlight reduced number of Tbx4 expressing cells in Pen^{-/-} mice being similar to loss of Pitx1^{-/-} and suggest that loss of Pitx1 expression by deletion of the enhancer might be mediated by Tbx4. Are the Tbx4-negative cells that result as a loss of Pen or Pitx1 transcriptionally similar? Can the authors show a UMAP plot of the scRNA-seq data to highlight the changes in the proportions of cells (to better compare and visualize the difference or similarities between the two genotypes)?
 - a. The authors' response to this comment is satisfactory.
6. Did the authors detect any interchromosomal contacts or other long-range contacts that might be altered in the

Pitx1^{-/-} and Pen^{-/-} models?

a. The authors include full chromosome 13 analysis, which is adequate to address this comment.

7. The cells that are able to maintain the GFP⁺ expression under loss of Pen maintain similar RNA profiles; which suggests there may also be other factors that contribute to the observed local chromatin conformation that supports active Pitx1 transcription. Can the authors speculate on what some of these factors might be?

a. The authors have included additional comments in the discussion to address this point, which is satisfactory.

Overall, the manuscript is substantially improved with suggestions provided by all reviewers.

We thank this reviewer for their positive assessment of our answers to their comments and suggestions.

Reviewer #4 (Remarks to the Author):

The authors have satisfactorily addressed my comments.

We thank this reviewer for their positive assessment of our answers to their comments and suggestions.